# Generating active T1 transitions through mechanochemical feedback

**Rastko Sknepnek[1,2]\*, Ilyas Djafer-Cherif[3†], Manli Chuai[2], Cornelis Weijer[2]\*, Silke Henkes[3,4]\***

[1]School of Science and Engineering, University of Dundee, Dundee, United Kingdom; [2]School of Life Sciences, University of Dundee, Dundee, United Kingdom; [3]School of Mathematics, University of Bristol, Bristol, United Kingdom; [4]Leiden Institute of Physics, Leiden University, Leiden, Netherlands

**Abstract** Convergence–extension in embryos is controlled by chemical and mechanical signalling. A key cellular process is the exchange of neighbours via T1 transitions. We propose and analyse a model with positive feedback between recruitment of myosin motors and mechanical tension in cell junctions. The model produces active T1 events, which act to elongate the tissue perpendicular to the main direction of tissue stress. Using an idealised tissue patch comprising several active cells embedded in a matrix of passive hexagonal cells, we identified an optimal range of mechanical stresses to trigger an active T1 event. We show that directed stresses also generate tension chains in a realistic patch made entirely of active cells of random shapes and leads to convergence–extension over a range of parameters. Our findings show that active intercalations can generate stress that activates T1 events in neighbouring cells, resulting in tension-dependent tissue reorganisation, in qualitative agreement with experiments on gastrulation in chick embryos.

**\*For correspondence:**
r.sknepnek@dundee.ac.uk (RS);
c.j.weijer@dundee.ac.uk (CW);
shenkes@lorentz.leidenuniv.nl
(SH)

**Present address:** [†]Dioscuri
Centre for Physics and Chemistry
of Bacteria, Institute of Physical
Chemistry, Polish Academy of
Sciences, Warsaw, Poland

**Competing interest:** The authors
declare that no competing
interests exist.

**Reviewing Editor:** Karsten
Kruse, University of Geneva,
Switzerland

## Editor's evaluation

This theoretical investigation provides important findings on the role of active mechanical feedback on tissue remodelling. The authors present convincing evidence that mechanically enforced myosin recruitment at cell-cell junctions can lead to tissue expansion in the direction perpendicular to an externally applied uniaxial mechanical stress. The relevance of the proposed mechanism for convergence–extension systems requires more investigation through comparison with experimental data.

## Introduction

Embryonic development involves complex tissue dynamics, including rearrangements and shape changes of the cells. This is particularly evident during gastrulation where the presumptive ectoderm, mesoderm, and endoderm take up their correct positions in the embryo (*Wolpert et al., 2015*). Key cellular processes that underlie tissue formation and morphogenesis during gastrulation are cell division, differentiation, and cell movement. Directed cell intercalation is a major mechanism driving large-scale tissue shape changes both in epithelial and mesenchymal tissues (*Huebner and Wallingford, 2018*). The narrowing and lengthening of epithelial tissues resulting from such intercalations, known as convergent extension (*Keller et al., 2000*), underlie germband extension in *Drosophila* (*Bertet et al., 2004*; *Blankenship et al., 2006*), as well as primitive streak formation in the chick embryo (*Voiculescu et al., 2007*; *Rozbicki et al., 2015*). In the latter, cell intercalations facilitate coordinated movements of hundreds of thousands of cells in two counter-rotating millimetre-scale cell flows that drive the formation of the primitive streak at the site where the flows meet (*Rozbicki et al., 2015*; *Saadaoui et al., 2020*; *Serrano Nájera and Weijer, 2020*). Unlike cell migration (*Alert*

*and Trepat, 2020*), which typically involves a significant contribution from crawling against a substrate such as the extracellular matrix, during intercalation, cells pull against each other in order to exchange their neighbours (*Huebner and Wallingford, 2018*). This is a complex, active process that requires a carefully coordinated shrinking and subsequent expansion of cell–cell interfaces, known as the T1 transition (*Kong et al., 2017*).

The widely studied morphological process of germband extension in *Drosophila* involves directed cell intercalations in the ventral ectoderm, where during the associated T1 events dorsal–ventral (DV)-oriented junctions shrink and new junctions are generated in anterior–posterior (AP) direction. The contraction of DV-oriented junctions has been shown to correlate with increased accumulation of apical myosin II in these junctions that can form supercellular cables in aligned junctions (*Bertet et al., 2004*; *Blankenship et al., 2006*). The extension of new junctions has been associated with the activity of medial myosin (*Rauzi et al., 2010*; *Collinet et al., 2015*). Laser ablation experiments in the ventral ectoderm have shown that the myosin-rich DV-oriented junctions are under higher tension than AP junctions (*Rauzi et al., 2008*; *Fernandez-Gonzalez et al., 2009*; *Collinet et al., 2015*), and aspiration and optical tweezing and optogenetic experiments have shown that myosin can be recruited to junctions in response to increased tension in these junctions, demonstrating the existence of mechanical feedback (*Fernandez-Gonzalez et al., 2009*; *Clément et al., 2017*; *Gustafson et al., 2022*). This is in agreement with observations in the *Drosophila* wing disk when it has recently been shown that myosin accumulates on apical junctions in response to mechanical stretching of the disk (*Duda et al., 2019*). It has recently been shown that the distribution of apical myosin can accurately predict the observed tissue flow patterns during germband extension (*Streichan et al., 2018*). However, an as yet unresolved question is by which mechanism the anisotropic distribution of myosin cables is initially generated. It is thought to depend on the family of Toll receptors under the control of pair rule genes (*Paré et al., 2014*) and their interactions with an adhesion G protein0coupled receptor (*Lavalou et al., 2021*) that could generate asymmetries in cells and can possibly signal to Rho-kinase and myosin; however, the precise molecular details remain to be resolved. Recently, a strong correlation between the DV junctional strain rate gradient and the junctional myosin recruitment rate gradient, both high at the ventral side, has been observed (*Gustafson et al., 2022*). This has led to the renewed suggestion that the myosin anisotropy may arise in response to extrinsic forces, such as those generated by the mesoderm invagination of the ventral furrow (*Butler et al., 2009*), a process that starts somewhat before germband extension and in combination with other extrinsic events such as posterior hindgut invagination, and other geometric constraints could drive germband extension (*Collinet et al., 2015*; *Gehrels et al., 2023*). In this scenario, the AP and DV patterning system could be involved in setting the level of mechanical feedback.

Experiments in the chick embryo showed that directed intercalations of mesendoderm precursors located in a sickle-shaped region in the posterior of the epiblast drive the tissue flows underlying the formation of the primitive streak (*Voiculescu et al., 2007*; *Rozbicki et al., 2015*). This sickle-shaped mesendoderm precursor region contracts along its long axis towards the AP midline of the embryo and extends along this midline in anterior direction to form the primitive streak. Measurements of the directions of intercalations show that they are aligned along the long axis of the mesendoderm sickle in the direction of the contraction of the sickle and correlate well with the direction and magnitude of the anisotropic strain rate component (*Rozbicki et al., 2015*; *Chuai et al., 2023*). These directed intercalations are mediated by super cellular myosin cables in aligned junctions in the direction of intercalation (*Rozbicki et al., 2015*; *Saadaoui et al., 2020*; *Serrano Nájera and Weijer, 2020*). The onset of tissue motion starts near the central midline of the sickle and then rapidly extends outwards to more lateral regions, suggesting the existence of an outward-propagating signal (*Rozbicki et al., 2015*). Furthermore, the intercalating cells go through a characteristic elongation in the direction of intercalation around the time of the onset of motion, which disappears when the epiblast tissue flows pickup speed. These observations, coupled with the fact that the chick embryo epiblast contains more than 60,000 cells at the onset of gastrulation requiring coordination of cell intercalation over large distances, led us to suggest that long-range mechanical signals coordinate the intercalations in the large scale (*Rozbicki et al., 2015*; *Serrano Nájera and Weijer, 2020*). More specifically, *Rozbicki et al., 2015*; *Serrano Nájera and Weijer, 2020* proposed that local contraction of a junction would, through an increase in tension in aligned junctions of neighbouring cells, activate a mechanical feedback process in those junctions, in turn, resulting in their contraction. This process could explain

the formation of the observed myosin cables in response to tension and result in coordinated and directed intercalations. So far, it has not yet been possible to test tension-dependent myosin recruitment directly in chick embryos since there has not been a live indicator of myosin activity. Currently, it is only possible to observe active myosin using a phospho-myosin light chain antibody in fixed embryos. The orientation and alignment of the myosin cables, however, correlate well with the anisotropic component of the strain rate tensor (*Rozbicki et al., 2015*; *Chuai et al., 2023*), making it likely that a tension-dependent recruitment of myosin occurs in chick embryos.

To develop a cell-level model of the convergence–extension process, it is necessary to understand how externally applied and internally generated mechanical stresses couple to the signalling pathways that regulate the cell's mechanical response. One, therefore, needs to understand the feedback between mechanical stress anisotropy and the anisotropy of the distribution of force-generating molecular motors in the cell, that is, how it emerges and is propagated and coordinated over large distances. The aim of this study is to formulate and analyse a model for cell intercalations that includes explicit mechanochemical coupling and does not require initial chemical prepatterning. The initial symmetry breaking is driven by anisotropic mechanical stresses rather than anisotropic distribution of signalling molecules. The focus of this work is on the mechanism of the T1 transition that occurs perpendicular to the direction of the maximum principal mechanical stress. We refer to such T1 transition that generate stress as *active*, which is different from T1 transitions that relieve stress by intercalating in a direction perpendicular to stresses generated by surrounding tissues. In the present model, this stress is assumed to be anisotropic and externally applied, while in an embryo it is produced by the tissue surrounding the region of interest, for example, the sickle-shaped region in the posterior of the chick embryo that develops into the primitive streak (*Serrano Nájera and Weijer, 2020*). Unlike passive T1 events that are local plastic rearrangements that relieve the applied stresses as, for example, in foams (*Weaire and Hutzler, 2001*), active T1 transitions studied here require the cell to induce junction contractions via self-amplifying generation of tension. The key ingredient of the model is, therefore, a feedback mechanism between the kinetics of the force-producing molecules, here assumed to be myosin, and mechanical tension in cell junctions. One of the simplest forms in which this feedback could be implemented is through the formation of well-documented catch bounds between myosin heads and actin filaments, which is highly relevant for proper muscle function (*Veigel et al., 2003*). In this catch-bond mechanism, the dissociation rate of myosin from actin filaments is a decaying exponential function of tension, where increased tension results in dissociation rate of myosin from the actin-cytoskeleton to be a simple exponential function of tension where increased tension results in a lower dissociation rate of myosin. Assuming that the association rate is not tension sensitive, this process will result in a net tension-dependent myosin accumulation.

Here we construct a model that generically provides a mechanism for active T1 events that underlie convergent extension flows such as those observed during primitive streak formation in the chick embryo (*Rozbicki et al., 2015*). Our analysis indicates that the viscoelastic nature of the cell–cell junctions is essential for an active T1 event, in agreement with studies on ratcheting during junction contractions (*Clément et al., 2017*; *Staddon et al., 2019*). In addition, for the active T1 transition to be possible, there must be a separation of elastic ($t^*$), viscoelastic remodelling ($\tau_{\mathrm{v}}$), and motor turnover timescales ($\tau_{\mathrm{m}}$), with $\tau_{\mathrm{v}}, \tau_{\mathrm{m}} > t^*$.

We first analyse the mechanochemical feedback in the case of a single junction, which describes the key ingredients of the proposed mechanism but avoids complications associated with cell rearrangements. The analysis then proceeds to the two-dimensional case, implemented as an extension of the vertex model (*Farhadifar et al., 2007*; *Fletcher et al., 2014*), where cell rearrangement is not only possible but leads to shape changes at the tissue scale. We find active T1 transitions and convergence–extension flows over a broad region of externally applied stresses and relaxation timescales, confirming that the proposed mechanism is robust.

## Results

### Single-junction model

To understand the mechanism that couples the kinetics of myosin motors to the local mechanical tension and leads to the activation of contractility in cell–cell junctions, we first analyse a model of a single junction. The single-junction model thus provides insight into the conditions under which

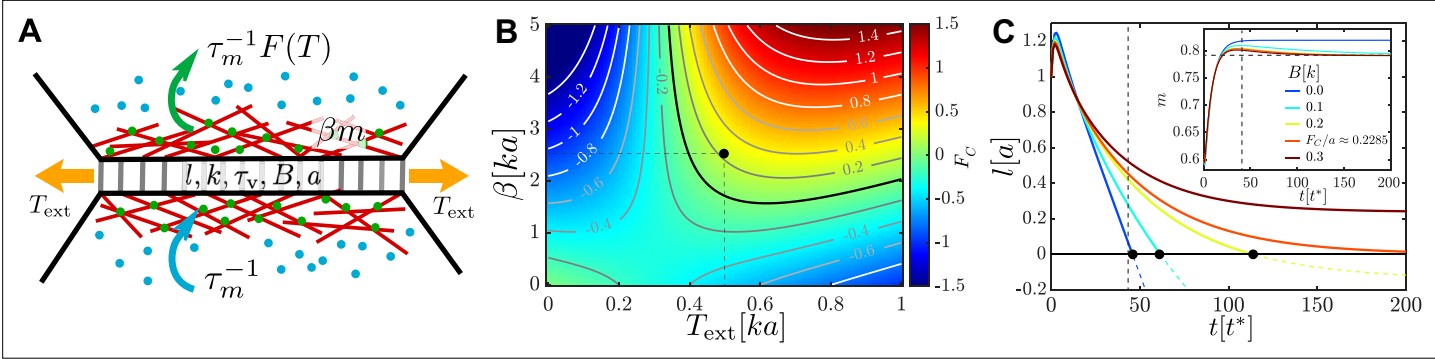

**Figure 1.** An active junction. (**A**) An external pulling force of magnitude $T_{ext}$ induces tension $T$ in a cell–cell junction of length $l$, which consists of passive viscoelastic and active components. The passive component consists of a Maxwell element with stiffness $k$ and viscous relaxation time $\tau_v$ and a harmonic spring of stiffness $B$ and rest length $a$ connected to it in parallel. The active component is due to myosin motors (green and blue dots) with concentration $m$ that act to contract cortical actin filaments (red lines), exerting a force of magnitude $\beta m$. Myosin motors bind to the actin cortex with association rate $\tau_m^{-1}$ and unbind with a tension-dependent dissociation rate $\tau_m^{-1}F(T)$. (**B**) Heatmap plot of the contraction force $F_C(T_{ext}, \beta)$. For $B = 0$, the junction contraction rate is $\dot{l} = F_C/\zeta$, where $\zeta$ is the friction coefficient with the surrounding medium. The mechanochemical feedback loop is contractile in the top-right quadrant where $\beta > \beta_c$, $T_{ext} > T^*$, and $F_C > 0$. Negative values of $F_C$ correspond to an extending junction. (**C**) Junction length vs. time for $T_{ext} = 0.5ka$, $\beta = 2.5ka$, $\tau_v = \tau_m = 10t^*$ (black dot in **B**) for increasing values of the elastic barrier $B$. An active T1 corresponds to reaching $l = 0$. Increasing $B$ slows down contractions, until, for $B \geq F_C/a$, the equilibrium length $l \geq 0$ and no T1 is possible. Inset: myosin dynamics for the same set of junctions; the horizontal dashed line indicates $m_{eq} \cdot \alpha = 1$, $T^* = 0.3ka$, $k_0 = 2/T^*$, and $m_0 = 0.5$. Length is measured in units of $a$, time in units of $t^* = \zeta/k$, and force in units of $ka$.

The online version of this article includes the following figure supplement(s) for figure 1:

**Figure supplement 1.** Key ingredients of the single active junction model.

**Figure supplement 2.** A linear chain of active junctions.

the junction length can contract to zero and trigger a T1 transition. In this model, for simplicity, the junction is assumed to be surrounded by a tissue which provides an elastic, tension-generating background against which it actively contracts. Guided by experiments, there are three key ingredients of the single-junction model. (1) The junction is viscoelastic, as established by pull-release optical tweezer experiments on cell–cell junctions in *Drosophila* and chick embryos (**Clément et al., 2017**; **Ferro et al., 2020**). This means that the junction is able to remove imposed tension by remodelling itself. (2) The junction can generate tension via the action of myosin motor minifilaments that slide actin filaments against each other. (3) There is self-amplifying feedback due to the exponentially decreasing unbinding rate of myosin motors with tension in the junction (**Veigel et al., 2003**; **Kovács et al., 2007**; **Figure 1A** and **Figure 1—figure supplement 1A and B**). Furthermore, epithelial tissues, and early-stage embryos, in particular, are under mechanical tension as revealed by tissue and cell junction cutting and tweezing experiments (**Clément et al., 2017**; **Ferro et al., 2020**). This tension generates an elastic background against which the junction contracts and expands. We refer to that background elasticity as the *elastic barrier* since it is assumed that it acts to prevent junction remodelling. In the full model, it additionally captures the yield stress of the underlying material that inhibits T1 transitions (**Bi et al., 2015**). Details of the model are discussed in 'Materials and methods', with parameter values given in **Table 1**.

In order to describe the three key ingredients that characterise a cell–cell junction, while taking into account the effects of the elastic background, we adopt a minimal description, that is, we assume that the junction consists of four elements connected in parallel: (1) a Maxwell element with stiffness $k$ and viscous relaxation timescale $\tau_v$, which models the viscoelastic character of the junction; (2) an elastic spring with spring constant $B$ and rest length $a$, which represents the elastic background, that is, the elastic barrier; (3) an active element that models the contribution of the cytoskeleton by generating active tension $\beta(m - m_0)$, where $\beta$ is the activity, $m$ is the ratio of the number of myosin motors bound to the junction and the maximum possible number of bound motors, and $m_0$ is the reference value of $m$; and (4) a dashpot with dissipation rate $1/\zeta$, which models dissipation with the surrounding medium. The first two elements form a standard linear solid (SLS) element (**Figure 1—figure supplement 1A**). The presence of $m_0$ in the active element is necessary to account for the possibility that active

**Table 1.** Values of the parameters in the single-junction model.
Units: length ($a$), time ($t^* = \zeta/k$), force ($ka$).

**Base**

| Parameter | Description | |
|---|---|---|
| $k$ | Spring constant | |
| $a$ | Barrier rest length | |
| $\zeta$ | Friction with substrate | |

**Model**

| Parameter | Description | Value range |
|---|---|---|
| $B$ | Barrier spring constant | $0 - 0.2k$ |
| $T_{\text{ext}}$ | Applied external tension | $0 - 1ka$ |
| $\beta$ | Myosin activity | $0 - 3ka$ |
| $\tau_{\text{v}}$ | Viscoelastic time | $10t^*$ |
| $\tau_{\text{m}}$ | Myosin time | $10t^*$ |
| $m_0$ | Myosin reference level | $0.5$ |
| $T^*$ | Threshold tension | $0.3ka$ |
| $k_0$ | Slope of $m$ vs. $T$ at $T^*$ | $2/T^*$ |
| $\alpha$ | Tension-independent myosin dissociation | $1$ |

contractions of the surrounding tissue are stronger than those in the junction, which would result in it expanding. Furthermore, since $m$ and $m_0$ measure the relative numbers of bound motors, the expression for the active force does not include the junction length (for further discussion, see 'Materials and methods'). Under these assumptions, the dynamics of a junction with length $l$ and the rest length $l_0$ is

$$\zeta \dot{l} = -T + T_{\text{ext}}, \quad \tau_{\text{v}} \dot{l}_0 = l - l_0, \quad \tau_{\text{m}} \dot{m} = 1 - mF(T), \tag{1}$$

where $T = k\left(l - l_0\right) + \beta\left(m - m_0\right) + B\left(l - a\right)$ is the junction tension, and $T_{\text{ext}}$ is external tension, which generates stresses in the junction. The feedback loop between the concentration of bound myosin motors and the mechanical tension is captured by the equation for the myosin dynamics, which incorporates a tension-independent myosin binding rate $\tau_{\text{m}}^{-1}$, and an unbinding rate $F(T)/\tau_{\text{m}}$ that decreases with tension as $F(T) = \alpha + e^{-k_0(T-T^*)}$. The third equation in *Equation 1*, therefore, describes a catch-bond-type mechanism (*Dembo et al., 1988*; *Veigel et al., 2003*; *Thomas et al., 2008*; *Prezhdo and Pereverzev, 2009*) for myosin kinetics. We remark that it is possible to construct different models for tension-dependent myosin kinetics, for example, with the motor binding rate being tension-dependent. This is, however, not expected to lead to qualitative differences. The key for the mechanism of the active T1 transition studied here is that there is increased contractile activity in response to external mechanical tension, without the need to specify its precise molecular origin. The reason, however, to consider tension-dependent unbinding is because the nonmuscle myosin II, which is the primary molecular motor that drives contractions in early embryonic tissues, is known to have actin association rates that are tension-independent, but exhibits tension-dependent dissociation (*Kovács et al., 2007*). At steady state $\dot{m} = 0$, and the equilibrium myosin $m_{\text{eq}} = F(T)^{-1}$ is a sigmoid function of tension. $T^*$, therefore, sets the threshold that separates low and high levels of attached myosin motors. $\alpha$ and $k_0$ are constants with choices of their values discussed in 'Materials and methods'.

The first two equations in (1) can be combined as $\zeta \dot{u} = -\frac{\zeta}{\tau_{\text{v}}}u - T + T_{\text{ext}}$, where $u = l - l_0$. The intersection of nullclines $\dot{u} = 0$ and $\dot{m} = 0$ defines the fixed points of the dynamics, $\left(m_{\text{eq}}, u_{\text{eq}}\right)$ (*Figure 1—figure supplement 1C*). Experiments of *Clément et al., 2017* showed that prolonged pulling forced the junction to remodel and retain the elongated shape. Therefore, the relevant regime consistent with observations in real tissues is where the viscoelastic remodelling and myosin association timescales are longer than the elastic relaxation timescale, that is, for $\tau_{\text{v}}, \tau_{\text{m}} > \zeta/k \equiv t^*$. For $B = 0$, there is a

unique stable fixed point $G0$ that determines the long-time dynamics of the junction and the length of the junction continues to change at a constant rate $\dot{l} = \dot{u} + \dot{l}_0 = u_{eq}/\tau_v$ (*Figure 1C*, the $B = 0$ curve). For a junction with no external load, therefore, a fixed point with $u_{eq} \leq 0$ corresponds to steady contraction, while a fixed point with $u_{eq} > 0$ corresponds to expansion.

In the presence of a finite elastic barrier of height $Ba$, the junction behaves as an elastic solid in the long-time limit, and it is in mechanical equilibrium with $T = T_{\text{ext}}$. This corresponds to a single steady-state solution of *Equations 1* at $u_{\text{eq}}^B = 0$, indicated by the fixed point GB (*Figure 1—figure supplement 1C*). The corresponding steady-state value of myosin,

$$m_{\text{eq}} = \frac{1}{\alpha + e^{k_0(T^* - T_{\text{ext}})}}, \tag{2}$$

is independent of $B$, reflecting the fact that in mechanical equilibrium external tension is balanced by the tension generated by myosin motors. The condition for a T1 transition to occur is, therefore, that the active tension due to myosin motors is sufficiently strong to shrink the junction to $l_0 = 0$. The mechanical equilibrium condition $T = T_{\text{ext}}$ at that point allows one to compute the magnitude of the contraction force $F_C$ the junction generates, or equivalently, the maximum barrier height, $F_C(T_{\text{ext}}, \beta) \equiv Ba = \beta(m_{\text{eq}} - m_0) - T_{\text{ext}}$ that a contracting junction can overcome. *Figure 1B* shows isolines of $F_C$, where positive values of $F_C$ correspond to junctions that can contract down to a T1 in the presence of a load, while negative values of $F_C$ correspond to junctions that cannot. Above a threshold in $T_{\text{ext}} \gtrsim T^*$ and $\beta > \beta_C$, the junction is able to gradually generate sufficiently large contraction forces required to overcome the elastic barrier and shrink down. Conversely, for $T_{ext} \lesssim T^*$ the junction expands, which is the appropriate regime for elongation after a T1. Therefore, there is positive feedback between mechanical tension and activity, which results from the assumption that the myosin association rate is independent of tension while the dissociation rate decays exponentially with it. The isoline $F_C = 0$ (*Figure 1B*, thick black curve), separates the contracting and the expanding regimes, and it corresponds to a critical threshold $\beta_c$ for a T1 transition,

$$\beta_c \geq \frac{T_{\text{ext}}}{m_{\text{eq}} - m_0}, \tag{3}$$

where $m_{\text{eq}}$ is given by *Equation 2*. *Figure 1C* shows the junction dynamics in the presence of barriers of different heights for $T_{\text{ext}} = 0.5ka$, $\beta = 2.5ka$ with $F_C \approx 0.2285ka$ (*Figure 1B*, black dot). The junction shrinks to a point for $B \leq F_C/a$, while for larger values of $B$ the contraction stops at finite $l$. The initial elongation of the junction (*Figure 1C*) is due to our choice of the initial value of $m$ and reflects the fact that it takes $\approx \tau_m$ for the active contractile machinery to kick in.

We conclude by emphasising the contractile response to applied tension of the single-junction model that will be at the heart of convergence–extension mechanism discussed below. That is, applying small external forces will lead to an expanding junction. Increasing the force, however, leads to the junction shrinking due to the increase of bound myosin motors. Once initiated, the process of activation can continue spontaneously. To showcase this, we applied a pulling force corresponding to $F_C = 0$ (neither contracting nor expanding) to a chain of active junctions connected in series (*Figure 1—figure supplement 2A*). An initial myosin pulse applied to the central junction causes it to contract. The contraction of the central junction activates contractions of the neighbouring junctions, leading to shrinking of the entire chain (*Figure 1—figure supplement 2B*). Finally, the contraction rate strongly depends on the timescale of viscoelastic relaxation in the junction. Both effects should be measurable experimentally.

## Vertex model with active junctions

The single-junction model serves as a building block for a model of the entire epithelial tissue. A natural way to proceed is to extend an existing model for tissue mechanics. Setting aside apicobasal polarity, which affects cell intercalations in real tissues (*Huebner and Wallingford, 2018*), it can be assumed that the mechanical properties of the epithelial tissue arise chiefly from the apical junction cortex, an approximation that is able to qualitatively capture many aspects of tissue mechanics (*Fletcher et al., 2014*; *Murisic et al., 2015*).

We, therefore, model the mechanical response of the tissue with the vertex model (*Farhadifar et al., 2007*; *Fletcher et al., 2014*). The appeal of the vertex model is that it is able to capture both

fluid and solid behaviours of epithelial tissues, including some aspects of the viscoelastic rheology (*Tong et al., 2021*; *Tong et al., 2022*). In the vertex model, the transition between the fluid state where cells can easily intercalate and tissue scale flows is possible, and the solid phase where intercalations are arrested is controlled by a single dimensionless geometric parameter, $p_0 = P_0/\sqrt{A_0}$ (*Staple et al., 2010*; *Bi et al., 2015*; *Park et al., 2015*; *Bi et al., 2016*).

Here we extend the vertex model to include activity via the mechanochemical coupling introduced for the single junction and define the vertex model with active junctions. Details of the model are given in 'Materials and methods'. Each junction, shared by two cells denoted as 1 and 2, is augmented by two active elements that model active contractions by the actomyosin cortex on either side. The tension in the junction is then

$$T = T^{\mathrm{P}} + k\left(l - l_0\right) + \beta_1\left(m_1 - m_0\right) + \beta_2\left(m_2 - m_0\right),\qquad(4)$$

where $T^{\mathrm{P}}$ is the passive contribution from the standard vertex model energy given in *Equation 11*. The junction rest length $l_0$ is, however, not constant but subject to viscoelastic relaxation with $\tau_{\mathrm{v}}\dot{l}_0 = l - l_0$. It is important to note that $T^{\mathrm{P}}$ depends only on the cell perimeter and not on junction length $l$ and, therefore, does not allow for anisotropic mechanical response to an applied anisotropic tension. This term is however responsible for the yield stress of the vertex model when attempting T1 transitions (*Bi et al., 2015*) and corresponds to the barrier term in the single-junction model. The additional spring term of the Maxwell element in *Equation 4* is, therefore, crucial for generating anisotropic tension. The myosin dynamics of each active element is coupled to a conserved myosin pool of size $M$ for each cell $C$ that determines the association rate of myosin motors to the junctions. As in the single-junction model, the myosin dissociation rate is modulated by mechanical tension, leading to the following equation for myosin kinetics:

$$\tau_{\mathrm{m}}\dot{m} = \left(M - m_{\mathrm{act}}^{\mathrm{C}}\right) - zmF\left(T\right) + \eta.\qquad(5)$$

Here, $m_{\mathrm{act}}^{\mathrm{C}} = \sum_{\mathrm{e}=1}^{z} m_{\mathrm{e}}$ is the total amount of activated myosin bound to the $z$ junctions of cell $C$, and we have included a noise component $\eta$ with zero mean and variance $f$ to model stochastic binding and unbinding of myosin. The overdamped dynamics of vertices is determined by force balance between friction, elastic forces due to deformations of the passive vertex model, and active forces due to pairs of active elements acting along the junctions connected to the vertex, that is, $\zeta\dot{\mathbf{r}}_{\mathrm{i}} = -\nabla_{\mathbf{r}_{\mathrm{i}}}E_{\mathrm{VM}} + \mathbf{F}_{\mathrm{i}}^{\mathrm{act}}$, where $\mathbf{r}_{\mathrm{i}}$ is the position of vertex $i$, $E_{\mathrm{VM}}$ is the energy of the passive vertex model (*Equation 9*), and the active term $\mathbf{F}_{i}^{\mathrm{act}}$ derives from the active elements introduced in *Equation 4*.

## Single active T1 transition in a hexagonal patch

We begin by discussing a single active T1 event (*Figure 2* and *Figure 3*). The unit of time is set by the elastic timescale $t^* = \zeta/\left(k + \Gamma\right)$, the unit of length by the side of a regular hexagon $a$, and the unit of force by $f^* = \left(\Gamma + k\right)a$, where $\Gamma$ is the perimeter modulus of the passive vertex model introduced in *Equation 9*. Values of the parameters used in simulations are listed in *Table 2*.

To understand the dynamics of a single active T1, we first studied a regular lattice of hexagonal cells that are passive except for an inclusion of four central active cells surrounded by a buffer ring at half activity (*Figure 2A*). The buffer cells were used to prevent distortions associated with large differences in activity between cells. Mechanical anisotropy was created by applying forces of equal magnitude and opposite direction perpendicular to the left and right boundaries to induce anisotropic mechanical stresses in the tissue. Furthermore, both activity and viscoelastic relaxation were switched off until mechanical equilibrium was reached (see 'Materials and methods'). The initial state is mechanically anisotropic (*Figure 2B*, top left), with differential tension in horizontal (h) vs. shoulder (s) junctions (*Figure 3A*), with $T_{\mathrm{h}}$ being significantly larger than $T_{\mathrm{s}}$. Near the equilibrium point $T = T^*$, and for $M = 6$, $m_{\mathrm{act}}^{\mathrm{C}} \approx 3$, the dynamics of the model tissue closely resembles the dynamics of the single-junction model. It is easy to show that $m_{\mathrm{eq}} \approx \left(2F(T)\right)^{-1}$, and mechanical anisotropy leads to anisotropic distribution of myosin, that is, $m_{\mathrm{h}} > m_{\mathrm{s}}$ (*Figure 2B*, bottom left). There is a range of applied pulling forces that, therefore, produce tensions $T_{\mathrm{s}} < T^* < T_{\mathrm{h}}$ in the system. In this regime, for a suitable $\beta > \beta_{\mathrm{c}}/2$ (*Equation 3*) horizontal junctions are contractile, while shoulder junctions are extensile. Here, the factor of 1/2 is due to both active elements of a junction acting in parallel.

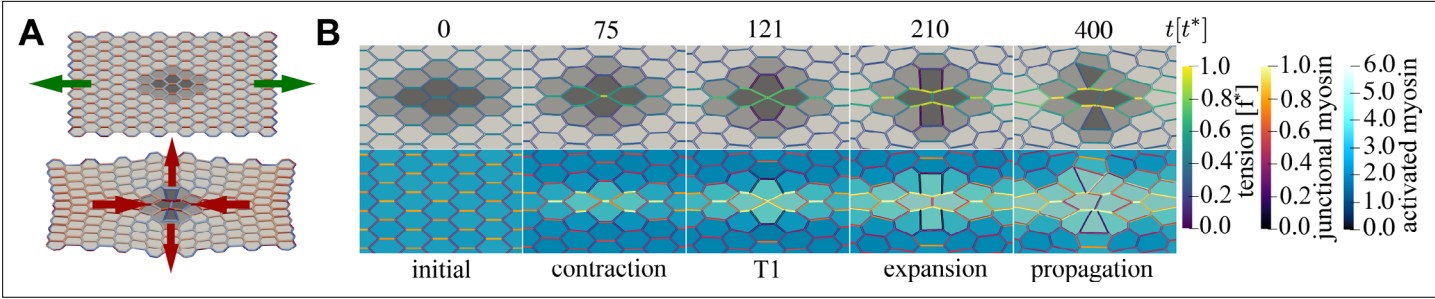

**Figure 2.** An active T1 transition event. (**A**) Top panel: the mechanical anisotropy in the initial state is produced by applying pulling forces (green arrows) in the horizontal direction to the left and right boundaries. Bottom panel: the final state after the active T1 shows a clear convergence–extension deformation (red arrows). Cells are coloured by type: passive (light grey), buffer (medium grey), and active (dark grey). Junctions are coloured by junctional myosin. (**B**) Time sequence of the active T1 transition measured from the moment activity and viscoelasticity were switched on. Cells in the top row are coloured by type and junctions are coloured by tension. Cells in the bottom row are coloured by activated myosin $m_{act}^C$, and junctions are coloured by myosin. Parameters: $A_0 = \frac{3}{2}\sqrt{3}a^2$, $P_0 = 6a$, $\beta = 0.8f^*$ (active), $\beta = 0.4f^*$ (buffer), $\beta = 0$ (passive), $M = 6$, $T^* = 0.3f^*$, $k_0 = 2/T^*$, $\tau_v = 20t^*$, $\tau_m = 100t^*$, $\alpha = 0.1$, $f = 1$, $f_{pull} = 0.15f^*$, with $n_x = 15$ ($n_y = 11$) cells in the horizontal (vertical) direction. Units: length ($a$), time ($t^* = \zeta/(\Gamma + k)$), force ($f^* = (\Gamma + k)a$).

The online version of this article includes the following video and figure supplement(s) for figure 2:

**Figure supplement 1.** Schematic representation of the key ingredients in the vertex model with active junctions.

**Figure 2—video 1.** Simulation of a passive system with applied horizontal pulling for $f_{pull} = 0.15f^*$.
https://elifesciences.org/articles/79862/figures#fig2video1

**Figure 2—video 2.** Low activity system with no T1 transitions.
https://elifesciences.org/articles/79862/figures#fig2video2

**Figure 2—video 3.** Active T1 transition.
https://elifesciences.org/articles/79862/figures#fig2video3

**Figure 2—video 4.** System with low pulling force.
https://elifesciences.org/articles/79862/figures#fig2video4

**Figure 2—video 5.** System with too high pulling force.
https://elifesciences.org/articles/79862/figures#fig2video5

**Figure 2—video 6.** System with too high activity.
https://elifesciences.org/articles/79862/figures#fig2video6

**Figure 2—video 7.** Active T1 transition with low $\tau_m$ and $\tau_v$.
https://elifesciences.org/articles/79862/figures#fig2video7

**Figure 2—video 8.** Active T1 transition with low $\tau_m$.
https://elifesciences.org/articles/79862/figures#fig2video8

**Figure 2—video 9.** Active T1 transition with low $\tau_v$.
https://elifesciences.org/articles/79862/figures#fig2video9

**Figure 2—video 10.** Active T1 transition at intermediate values of $\tau_m$ and $\tau_v$.
https://elifesciences.org/articles/79862/figures#fig2video10

**Figure 2—video 11.** Active T1 transition at high $\tau_v$.
https://elifesciences.org/articles/79862/figures#fig2video11

**Figure 2—video 12.** Active T1 transition at high $\tau_m$.
https://elifesciences.org/articles/79862/figures#fig2video12

**Figure 2—video 13.** Active T1 transition at high $\tau_m$ and $\tau_v$.
https://elifesciences.org/articles/79862/figures#fig2video13

In steady state, both activity $\beta$ and viscoelasticity were switched on (*Figure 2—videos 1–6*), with timescales chosen such that $\tau_v, \tau_m \gtrsim t^*$, which is the biologically relevant regime. *Figure 2B*, second column, shows the system after $75t^*$. For the central horizontal active junction, contractility has been triggered, and the junction is steadily contracting at high myosin and against high tension (*Figure 3B*). The active T1 transition is reached at $121t^*$, when the central junction shrinks to a point and a four-vertex is created (*Figure 2B*, third column). If the sum of the forces on the four-vertex is favourable for

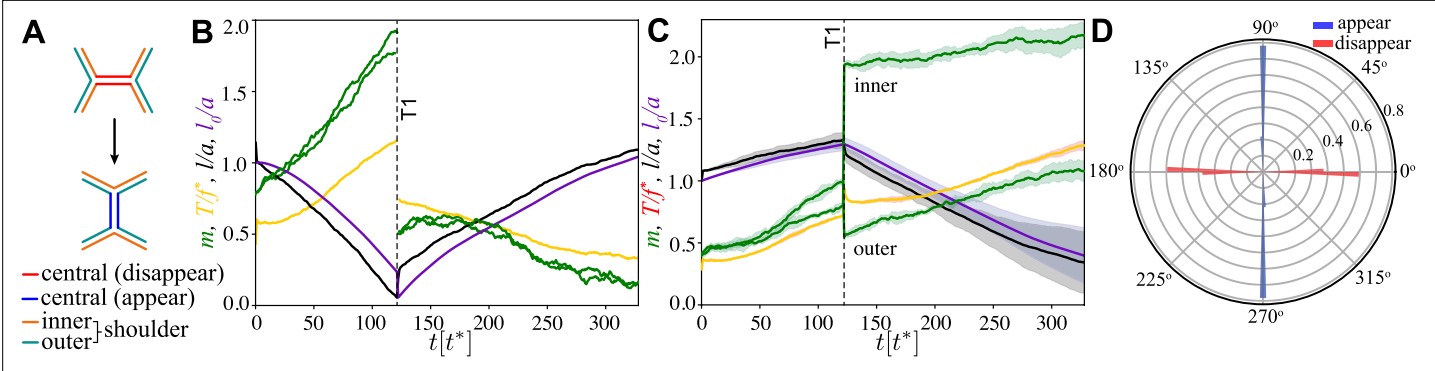

**Figure 3.** Junction dynamics during the active T1 transition shown in *Figure 2*. (**A**) Definition of central (red/blue – junction that disappears/appears), inner (orange), and outer (blue-green) shoulder junctions through the T1 transition. (**B**) Central junction: myosin, $m$ (green; two curves for myosin on two sides of the junction), tension, $T$ (yellow), junction length, $l$ (black), and rest length $l_0$ (purple) vs. time. The vertical line indicates the T1 transition, at which point junctional myosin is redistributed according to the rules outlined in *Figure 2—figure supplement 1B*. (**C**) Same as in panel (**A**) but averaged over four shoulder junctions, with variance indicated as shade. (**D**) Polar histogram of the orientation of the first T1 event measured with respect to the pulling direction, from $n = 32$ simulations. Blue (red) indicates appearing (disappearing) junctions. Parameters are $\beta = 0.8f^*$ and $f_{\text{pull}} = 0.15f^*$ and as in *Figure 2*.

The online version of this article includes the following figure supplement(s) for figure 3:

**Figure supplement 1.** Continuous strain tensors through the active T1 transition, for $\beta = 0.8f^*$, $f_{\text{pull}} = 0.15f^*$, and the other parameters corresponding to *Figure 2* and *Figure 3*.

it to split (*Spencer et al., 2017*) in the vertical direction, a T1 event occurs, accompanied by myosin redistribution (see 'Materials and methods'). The newly created vertical junction expands at intermediate values of myosin and tension (*Figure 2B*, fourth column, and *Figure 3B*). In contrast, aided by the anisotropic redistribution of myosin, the shoulder junctions are now strongly polarised and begin contracting, with higher myosin levels on the side of the junction belonging to the expanding pair of cells (*Figure 3C*). This expansion phase is followed by several secondary T1 events, the first of which typically occurs at a shoulder junction (*Figure 2B*, fifth column). During this propagation phase, there is a strong mechanical anisotropy in the direction of applied pulling forces. Together, the central T1 and the subsequent T1 events lead to substantial convergence–extension flow as can be seen qualitatively in the shape of the region formed by the 14 central and buffer cells (*Figure 2*, medium and dark grey).

## Timescales of active T1 events and local convergence–extension strain

We proceed to quantify T1 transitions using the method introduced by *Graner et al., 2008* (for a brief summary, see 'Materials and methods'). First, using the topological tensor, $\hat{\mathbf{T}}$ (*Equation 19*), we measured the time and orientation of the T1 transition along the direction of the central junction of the active region (hereafter the 'central T1'). *Figure 4A* shows the probability of a central T1 as a function of $f_{\text{pull}}$ and $\beta$, with other parameters held constant at the same values as in *Figure 2*. The probability was computed from $n = 32$ simulations with different realisations of the myosin noise as the fraction of simulations where the first observed T1 was along the central active junction rather than elsewhere in the system. The probability of any T1 in the system was also measured, with the red contour in *Figure 4A* corresponding to 50% of realisations having a T1.

These results show that there is an absolute lower threshold, $\beta > \beta_c/2$, for any form of T1 to occur. This is qualitatively consistent with both sides of the junction acting as two parallel instances of the single-junction model. Second, there is an optimal range of applied pulling forces for central T1 transitions, $0.1 < f_{\text{pull}}/f^* < 0.2$. Within this range, $T_s < T^* < T_h$ for the central and shoulder junctions, and during the initial contracting phase, they are contractile and extensile, respectively. Outside this optimal regime, the probability for central T1 events decreases rapidly, though T1 transitions still occur elsewhere for large values of $\beta$.

The orientation of the central T1 transition in the optimal region is shown in *Figure 3D*. It was computed from the angle of the principal direction of $\hat{\mathbf{T}}$ corresponding to the largest eigenvalue before and after the T1. One can immediately observe that the transition is highly symmetric, with the

**Table 2.** Values of the parameters used in the vertex model with active junctions.
Units: length ($a$), time ($t^* = \zeta / (\Gamma + k)$), force ($f^* = (\Gamma + k)\, a$).

**Base**

| Parameter | Description | |
|---|---|---|
| $a$ | Hexagonal cell edge length | |
| $\Gamma$ | Perimeter modulus | |
| $k$ | Spring constant | |
| $\zeta$ | Friction with substrate | |

**Model**

| Parameter | Description | Value range |
|---|---|---|
| $\kappa$ | Area modulus | $1 f^*/a^3$ |
| $A_0$ | Target cell area | $3\sqrt{3}a^2/2$ |
| $P_0$ | Target cell perimeter | $6a$ |
| $f_{\text{pull}}$ | Pulling force | $0.0 - 0.3 f^*$ |
| $\beta$ | Myosin activity | $0.0 - 1.4 f^*$ |
| $\tau_{\text{v}}$ | Viscoelastic time | $10^0 - 10^3 t^*$ |
| $\tau_{\text{m}}$ | Myosin time | $10^1 - 10^3 t^*$ |
| $T^*$ | Threshold tension | $0.3 f^*$ |
| $K_0$ | Slope of $m$ vs. $T$ at $T^*$ | $2/T^*$ |
| $m_0$ | Myosin reference level | $0.5$ |
| $M$ | Total cell myosin | $6$ |
| $f$ | Variance of myosin fluctuations | $1$ |
| $\alpha$ | Tension-independent myosin dissociation | $0.1$ |

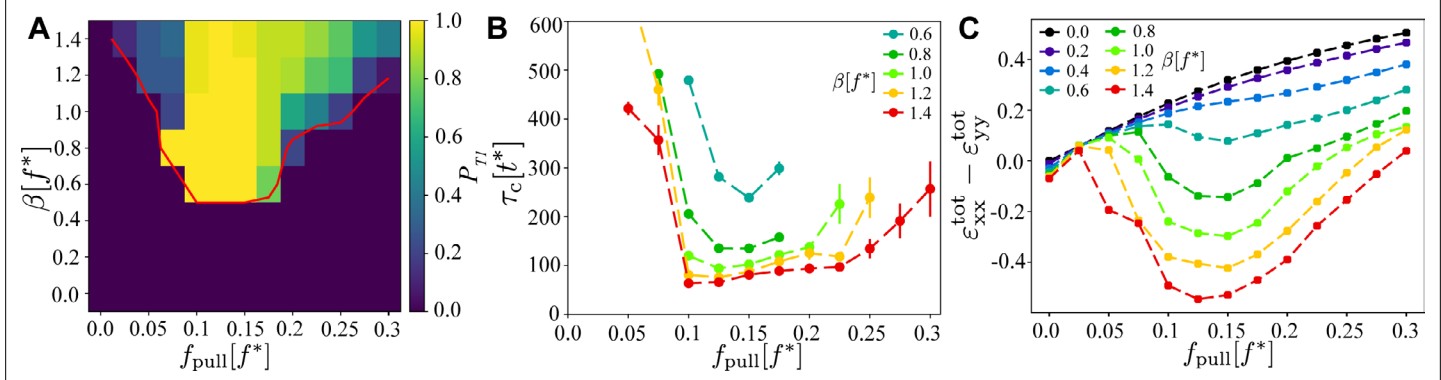

**Figure 4.** Existence and timescales of T1 transitions in the vertex model with active junctions as a function of $f_{\text{pull}}$ and $\beta$, averaged over $n = 32$ simulations with different realisations of the myosin noise. (**A**) Probability of a central T1 transition. The red line is the 50% probability contour of any T1 occurring in the simulation. (**B**) Typical timescale for the T1 transition to occur, measured as the length of the contraction phase. The other parameters are the same as in *Figure 2*. (**C**) Magnitude of the convergence–extension deformation as a function of $f_{\text{pull}}$ and characterised by measuring $\varepsilon_{\text{xx}}^{\text{tot}} - \varepsilon_{\text{yy}}^{\text{tot}}$ induced by the T1 transition.

orientation of the shrinking (disappearing) junction being very close to horizontal, and the expanding (appearing) junction very close to vertical. Outside the optimal regime, this symmetry disappears, with T1 events of non-central junctions occurring in different directions, similar to the fully random case discussed below.

*Figure 4B* shows the time to the first *T1* transition, $\tau_c$, measured from the point when the activity and viscoelastic relaxation were switched on. The analysis was limited to central T1 transitions at parameter values where at least 25% of simulations yield a central T1, with $\tau_v = 20t^*$ and $\tau_m = 100t^*$. One immediately observes that $\tau_c \gtrsim \tau_v, \tau_m$, consistent with a T1 dynamics being dominated by myosin activation and viscoelastic relaxation. We find that $\tau_c$ has a minimum in the same optimal region identified in *Figure 4A*, with $\tau_c$ rising both for larger and smaller values of $f_{pull}$. Furthermore, increasing $\beta$ beyond $\beta_c = 0.6f^*$ gradually reduces $\tau_c$, consistent with active contractions becoming stronger. For $\beta > 1.0f^*$, cells shapes become increasingly distorted, suggesting that the model is no longer applicable.

We now quantify convergence–extension generated by the model. The shear component of the total integrated strain $\varepsilon_{xx}^{tot} - \varepsilon_{yy}^{tot}$ given in *Equation 24* was computed for the 14 cells comprising the central and buffer regions (*Figure 2A*). *Figure 4C* shows $\varepsilon_{xx}^{tot} - \varepsilon_{yy}^{tot}$ as a function of $f_{pull}$, evaluated at $t = 400t^*$ and averaged over $10t^*$. This point approximately corresponds to the empirically determined peak of convergence–extension in the optimal region of applied pulling forces (see sample time traces in *Figure 3—figure supplement 1*). From *Figure 4C* it is evident that the model generates pronounced convergence–extension. Without activity, that is, for $\beta = 0$, the system extends in the direction of the applied pulling force and contracts perpendicular to it with $\varepsilon_{xx}^{tot} - \varepsilon_{yy}^{tot} > 0$ and, as expected, it increases with $f_{pull}$. For $\beta > 0$, $\varepsilon_{xx}^{tot} - \varepsilon_{yy}^{tot} > 0$ decreases, indicating that the activity acts against the applied pulling force. As $\beta$ increases beyond a critical value, $\beta_c \approx 0.6f^*$, active forces are strong enough to counteract the pulling forces and the system shrinks against the external load and extends in the perpendicular direction, that is, $\varepsilon_{xx}^{tot} - \varepsilon_{yy}^{tot} < 0$. During this process there are no significant changes of the area, that is, $\varepsilon_{xx}^{tot} + \varepsilon_{yy}^{tot} \approx 0$. This mechanism is, however, effective only for a range of values of $f_{pull}$. If $f_{pull}$ is insufficiently strong, the myosin–tension feedback loop does not fully activate (*Figure 2—video 4*). Conversely, if $f_{pull}$ is too strong, the feedback loop is active, but all junctions are activated, stiffening the tissue (*Figure 2—video 5*).

We conclude the analysis of a hexagonal tissue patch by investigating the influence of $\tau_v$ and $\tau_m$ (*Figure 2—videos 7–13*). *Figure 5A* shows the probability of a central T1 for $\beta = 1.0f^*$, $f_{pull} = 0.15f^*$, that is, deep in the optimal region, as a function of $\tau_m$ and $\tau_v$. Here, we only consider the biologically plausible regime with $\tau_m, \tau_v \gtrsim t^*$, with proportionally scaled myosin noise $f = 1$, and we excluded very small values of $\tau_m$ where noise dominates. We find that the mechanism for T1 events is very robust over 2–3 orders of magnitude in both $\tau_m$ and $\tau_v$, with a guaranteed T1 transition in most of the parameter space. The only exception is the regime $\tau_m \gtrsim 10\tau_v$, that is, very slow myosin dynamics compared

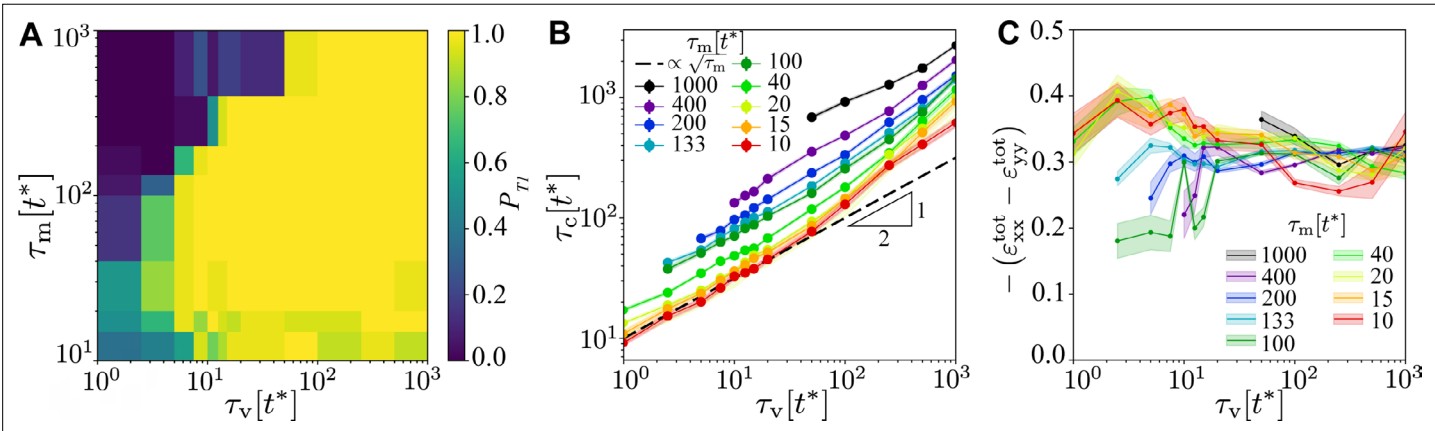

**Figure 5.** Robustness of the T1 mechanism as a function of $\tau_v$ and $\tau_m$ for $\beta = 1.0f^*$ and $f_{pull} = 0.15f^*$. (**A**) Probability of a central T1 event, averaged over $n = 32$ simulations with different realisations of the myosin noise. The probability of any T1 event is 1 throughout. (**B**) Contraction time to collapse for the central T1 as a function of $\tau_v$, for different values of $\tau_m$, for points where a central T1 event occurred in at least 25% of simulations. (**C**) Peak of the total convergence extension strain $\varepsilon_{xx}^{tot} - \varepsilon_{yy}^{tot}$, showing very weak dependence on viscoelastic and myosin timescales. Shading in panels (**B**) and (**C**) indicates the standard error of the mean.

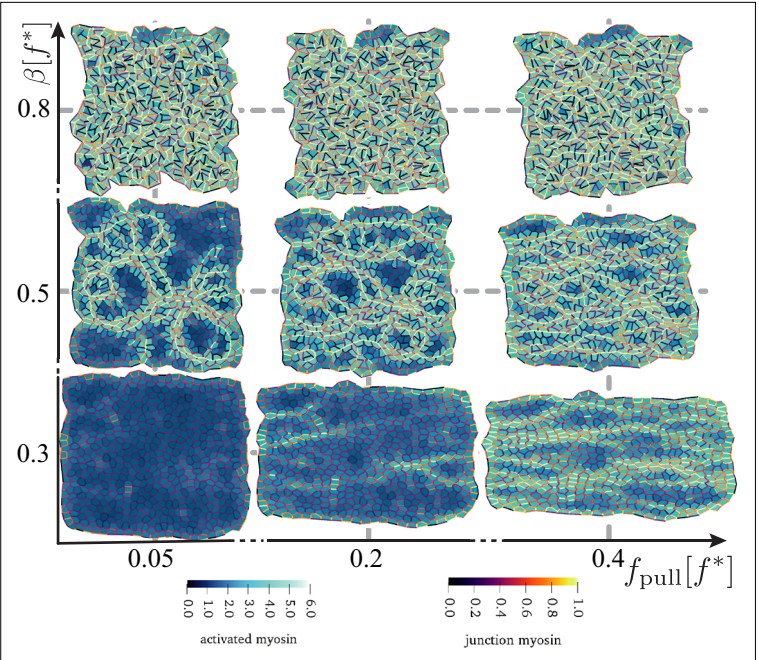

**Figure 6.** Disordered active tissue at time $t \approx 700t^*$ as a function of the magnitude of the pulling force $f_{\text{pull}}$ and activity $\beta$. The region of convergence–extension is at the centre of the diagram, around $\beta \approx 0.4 - 0.6f^*$ and $f_{\text{pull}} \approx 0.1 - 0.3f^*$. The remaining parameters are the same as in **Figure 2**.

The online version of this article includes the following video(s) for figure 6:

**Figure 6—video 1.** Video of simulations of active contractions in disordered tissues for a range of activities, β, and magnitudes of the pulling force, $f_{\text{pull}}$.

https://elifesciences.org/articles/79862/figures#fig6video1

**Figure 6—video 2.** Active T1 transitons in a random patch.

https://elifesciences.org/articles/79862/figures#fig6video2

to viscous relaxation, where the system fails to develop anisotropy. All simulations generated at least one T1, and there is no equivalent of the red contour in **Figure 4A**. In **Figure 5B**, we show the timescale of the T1 transition. We find the T1 timescales as $\tau_c \sim \tau_v^{1/2}$ and $\tau_c \sim \tau_m^{1/3}$. This influence of both timescales is consistent with the complex interplay between myosin activation on central and shoulder junctions. Finally, in **Figure 5C**, we show the convergence–extension strain $\varepsilon_{\text{xx}}^{\text{tot}} - \varepsilon_{\text{yy}}^{\text{tot}}$ as a function of $\tau_v$ for a range of values of $\tau_m$. The effectiveness of the T1 mechanism is largely independent of $\tau_v$ and $\tau_m$, and we have $\varepsilon_{\text{xx}}^{\text{tot}} - \varepsilon_{\text{yy}}^{\text{tot}} \approx -0.3$, the same value as in **Figure 4C**. This additional robustness of the model to varying timescales is likely due to being in a quasistatic regime where the elastic deformation timescale is much shorter than any other timescale.

## Convergence–extension in a fully active random patch

The hexagonal tissue patch is convenient to analyse isolated active T1 events. Cells in real epithelia, however, do not have regular shapes packed in crystalline order. We, therefore, studied a patch of 520 active and 80 passive cells generated from a centroidal Voronoi tessellation starting from $N = 600$ points placed at random in the simulation box. As in the case of the hexagonal patch, we applied pulling forces of constant magnitude on left and right boundaries in order to generate internal anisotropic stresses that mimic the mechanical conditions in the sickle region of the embryo (see next section). First, we investigated the occurrence of active T1 transitions and the emergence of convergence–extension as a function of $f_{\text{pull}}$ and $\beta$, at fixed $\tau_v = 20t^*$ and $\tau_m = 100t^*$, that is, in the same region of parameter space as in **Figure 4** (see also **Figure 6**). To be consistent with the hexagonal patch, the unit of length here is set by the length of a regular hexagon of area $L^2/N$, where $L$ is the initial patch size (see 'Materials and methods').

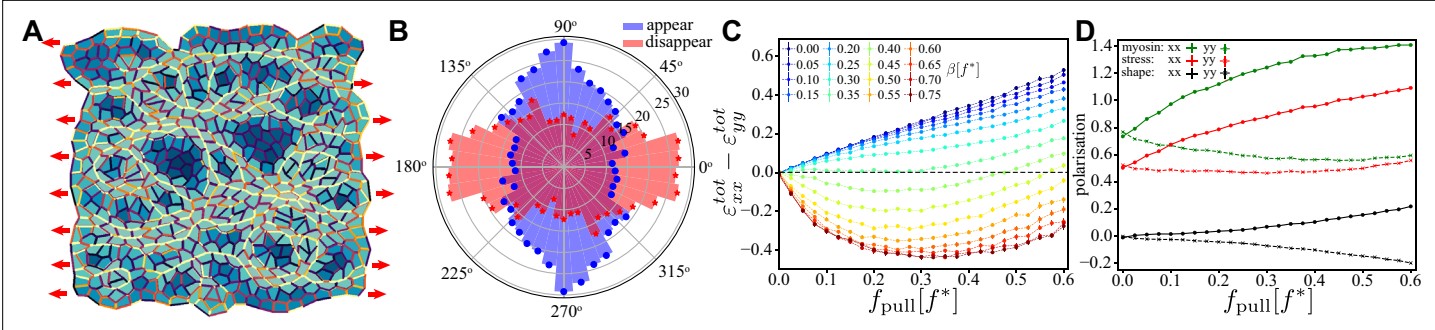

**Figure 7.** Characterisation of convergence-extension in a random tissue patch. (**A**) Snapshot of a random tissue patch for $\beta = 0.5f^*$ and $f_{\text{pull}} = 0.2f^*$ at $t \approx 700t^*$, that is during the convergence–extension flow. Red arrow indicates that a constant pulling force is applied throughout the entire simulation. (**B**) Angular histogram of T1 events for same values of $\beta$ and $f_{\text{pull}}$. Cells and junctions are coloured as in **Figure 6**. (**C**) Magnitude of the convergence–extension deformation as a function of $f_{\text{pull}}$ characterised by measuring $\epsilon_{xx}^{\text{tot}} - \epsilon_{yy}^{\text{tot}}$ induced by the T1 transition. (**D**) Anisotropy along ($xx$, solid line) and perpendicular to ($yy$, dashed line) the direction of the external pulling force for myosin (green), mechanical stress (red), and shape tensor (black) as functions of $f_{\text{pull}}$ for $\beta = 0.5f^*$ at $t \approx 700t^*$. In (**C**) and (**D**), each point was averaged over $n = 33$ independent samples and the error bar is smaller than the symbol size.

The online version of this article includes the following figure supplement(s) for figure 7:

**Figure supplement 1.** Continuous strain tensors for the fully active tissue at $\beta = 0.5f^*$, $f_{\text{pull}} = 0.2f^*$, and the other parameters corresponding to **Figure 2** and **Figure 3**.

**Figure supplement 2.** Measuring T1 transitions in the fully active random patch.

**Figure supplement 3.** T1 histograms in the parameter range where no convergence–extension occurs.

We observed T1 transitions for all simulated systems, though their rate rapidly increased when either $\beta$ or $f_{\text{pull}}$ were increased. Unlike in the case of the active inclusion in a hexagonal passive patch, there is no clear threshold for active T1 transitions. Instead, parts of the tissue are activated, and we observed the emergence of pronounced myosin cables and accompanying tension chains (e.g. **Figure 6**, middle row). The system starts to experience significant flow and convergence–extension from $\beta = 0.4f^*$, significantly below the values observed in the hexagonal patch with a single active inclusion where, depending on the magnitude of the applied force, T1 events start to appear for $\beta$ above $0.6 - 1.0f^*$. This suggests cooperative rearrangements in the tissue, and we indeed see evidence of serial active T1 transitions along tension chains (**Figure 6—video 2**). This suggests that the individual junction feedback mechanism together with mechanical (as opposed to chemical) propagation of myosin activation is a key ingredient in the formation of the myosin cables that have been observed to accompany convergence–extension flow in chick embryo gastrulation (**Rozbicki et al., 2015**) and in *Drosophila* germ band extension (**Jacinto et al., 2002**).

The system starts to rearrange from $\beta \approx 0.4f^*$, and for $\beta \gtrsim 0.7f^*$, one observes a highly active state with many uncorrelated rearrangements and implausible cell shapes. The region with realistic cell shapes is significantly below the hexagonal patch with a single active inclusion, where, depending on the magnitude of the applied force, T1 events start to appear for $\beta$ above $0.6 - 1.0f^*$.

In **Figure 7A**, we show a snapshot of a random patch long after ($\approx 700t^*$) activity was switched on for $\beta = 0.5f^*$ and $f_{\text{pull}} = 0.2f^*$, in what emerges to be the optimal region for convergence–extension. Oriented myosin cables accompanied by tension chains form in the initial stage of the simulation at $\approx 100t^*$ after activity was turned on. At longer times, the pronounced orientation of myosin and tension decreases (but does not disappear entirely), and convergence–extension stops (see 'Materials and methods' for details). The convergence–extension process is accompanied by active T1 events that predominately occur perpendicular to the direction of the external pulling, as shown in the orientational histogram (**Figure 7B**).

**Figure 7C** quantifies the amount of convergence–extension by measuring the difference of total integrated strain in directions along and perpendicular to the direction of the applied pulling force, that is, $\epsilon_{xx}^{\text{tot}} - \epsilon_{yy}^{\text{tot}}$. It is evident that like the hexagonal case, the random patch undergoes substantial convergence–extension over a range of activities. The process in accompanied by spatially anisotropic distribution of myosin, mechanical stress, and cell shapes (**Figure 7D**), which we measured through

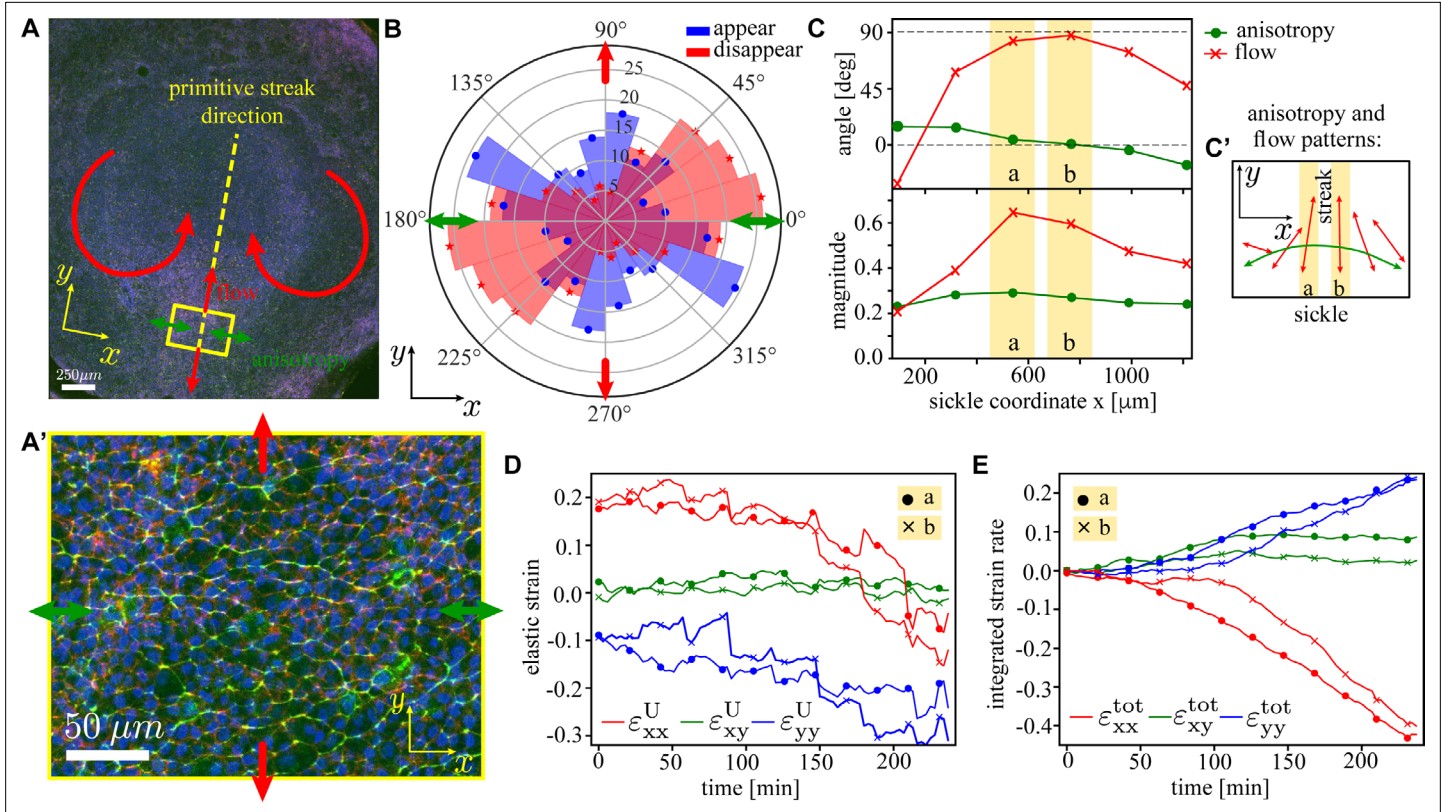

**Figure 8.** Analysis of the tissue flows in the early-stage chick embryo. (**A**) Image of a typical early-stage chick embryo prior to the gastrulation (i.e. primitive streak formation). The primitive streak will form along the yellow dashed line. The direction of myosin anisotropy is shown by the green double-headed arrows, and the direction of the tissue flow is indicated by the red arrows. *x*-axis is chosen to coincide with the long direction of the sickle-shaped active region in the embryo's posterior (*Rozbicki et al., 2015*). (**A'**) Zoom-in of the rectangular region on the posterior side of the embryo; myosin II (green), actin (red), and nuclei (blue). (**B**) Measured distribution of the orientation of T1 events in a circular patch of diameter ≈ 190 $\mu m$ tracked over the period of ≈ 6 hr (cf. model distribution in *Figure 7B*). Blue (red) denotes junctions that appear (disappear); arrows have the same meaning as in (**A**). (**C**) Angle (red) and magnitude (green) of tissue shape anisotropy (dots) and tissue flow (crosses) for $n = 6$ rectangular patches along the sickle with corresponding anisotropy and flow patterns shown in (**C'**). Components of the elastic strain tensor $\hat{\mathbf{U}}$ (**D**) and the total integrated strain tensor $\hat{\mathbf{V}}$ (**E**; definition in *Equation 24*) as a function of time during first 4 hr of the streak formation for two central regions of the sickle (yellow stripes in **C**). Details of the analysis are given in 'Materials and methods'.

The online version of this article includes the following video and figure supplement(s) for figure 8:

**Figure supplement 1.** Region of interest in the anterior of the embryo.

**Figure 8—video 1.** Image sequence taken at the base of the forming streak showing cell intercalations.

https://elifesciences.org/articles/79862/figures#fig8video1

**Figure 8—video 2.** Image sequence taken in the region of epiblast in front of forming streak.

https://elifesciences.org/articles/79862/figures#fig8video2

the eigenvalues of the myosin (*Equation 26*) and tension (*Equation 25*) tensors, and the eigenvalues of the shape tensor (*Equation 20*).

## Comparison with myosin-driven intercalations observed in the sickle-shaped mesoderm precursor domain of chick embryo at the onset of gastrulation

Previous experiments have shown that the embryo scale tissue flows driving the lateral to medial convergence and posterior anterior extension of the posterior sickle-shaped mesendoderm precursor domain that results in its transformation into the streak is driven by directed cell intercalations. These cells undergoing directed intercalations are characterised by an anisotropic cell shape and an anisotropic distribution of active myosin II in their cell junctions, organised in supercellular chains of variable

length (*Figure 8A and A'*; *Rozbicki et al., 2015*; *Chuai et al., 2023*). We here make a qualitative comparison of the myosin-driven intercalations in the model with the myosin-dependent directed cell intercalations driving tissue deformations in the sickle-shaped mesendoderm precursor region of the early gastrulation stage chick embryo. *Figure 8A* shows an image of a typical embryo prior to primitive streak formation with a patch magnified in panel A'. Both the myosin (green) and cell shapes exhibit a clear anisotropy (indicated by green arrows) along the lateral–medial axis of the mesendoderm precursor domain. The myosin cables are believed to reflect and generate the tension chains observed in simulations.

We have quantified the experimental tissue convergence–extension as follows. *Figure 8B* shows an orientational histogram of T1 events in a rectangular region of size $\approx 200 \times 200 \; \mu\mathrm{m}^2$ along the sickle tracked over a period of approximately 6 hr. The events were identified by calculating the $\hat{\mathbf{T}}$ tensor using segmented images (see 'Materials and methods' and *Figure 8—video 1*), and validated and corrected manually. In *Figure 8C*, we analysed $n = 6$ tissue patches of diameter $\approx$ 190 µmm chosen sequentially along the sickle-shaped region. We computed tensors that measure shape anisotropy $\hat{\mathbf{U}}$ and strain rate $\hat{\mathbf{V}}$ (*Graner et al., 2008*) by tracking the patches along the tissue flow over approximately 4 hr ('Materials and methods'). The anisotropy remains constant at around 20% and its direction is at the angle close to 90°, that is, along the sickle and orthogonal to the streak (*Figure 8C'*, green arrow). We also computed the mean flow magnitude and direction from the eigenvalues and eigenvectors of the integrated $\hat{\mathbf{V}}$ tensor. There is a pronounced spatial pattern to its direction, pointing towards the streak and parallel to the direction of anisotropy on outer parts of the sickle, and orthogonal to the sickle and its orientation in the middle of the sickle (*Figure 8C'*, red arrows). At the same time, the magnitude of flow peaks in the middle part of the sickle. This is consistent with the incipient flow to create the streak. In *Figure 8D and E*, we show the time dependence of spatial anisotropy and total strain, for two central patches. We see that anisotropy is along $x$ (i.e. perpendicular to the streak) and flow is along $y$ (i.e. along the streak), corresponding to convergence–extension flow. For comparison, we show the behaviour in an isotropic region anterior to the streak that shows non-directional intercalations and absence of tissue deformation (*Figure 8—figure supplement 1*, *Figure 7—figure supplement 3C*, and *Figure 8—video 2*). These observations show a close correspondence with the results of the model in the absence of imposed stress anisotropy, where the myosin-mediated active intercalations emerge spontaneously, leading to disorganised tissue patches with random intercalation directions (shown in *Figure 7—figure supplement 3B*).

## Discussion

There are two key features of our model with active junctions. First, the anisotropy of myosin distribution is induced by anisotropic mechanical tension. Second, active contractions are triggered by tension-sensitive accumulation. The feedback loop between tension and myosin motor activity leads to contraction against and extension perpendicular to tension. This cellular mechanism has been suggested to be driving the tissue flows during primitive streak formation in avian embryos (*Rozbicki et al., 2015*), where there is no clear evidence of chemical prepatterning. Although it is yet to be experimentally confirmed, it is plausible that the symmetry breaking event that induces the initial myosin anisotropy occurs as a result of anisotropic tension combined with cell differentiation early in development. For example, in the chick embryo, *Gdf3* is expressed in the sickle-shaped region in the posterior epiblast, and it is believed to play a role in triggering the contractions that initiate the large-scale tissue flows that subsequently lead to the formation of the primitive streak (*Serrano Nájera and Weijer, 2020*). The key conclusion of this study is that once the process has been initiated, chemical anisotropy emerges spontaneously and there is no need to impose it.

Although the model investigated here generates active T1 events, the process loses coherence after several T1 events, resulting in biologically implausible tissue shapes. This is very prominent in the toy case of regular hexagonal patch. While the problem is to some extent alleviated in patches of randomly shaped active cells, generating robust convergence–extension flows that would span scales of the entire embryo will be hard to achieve with the current model. One source of instability is likely that the post-T1 expansion of the junction, currently effectively modelled as a passive process, is not properly captured by the model and requires additional sources of activity to be considered. At the scale of the entire embryo, other cellular processes such as cell division, differentiation, and ingression all play non-trivial roles. These events have not been considered here.

Furthermore, recent results confirm that the mechanics of vertex models is complex (*Bi et al., 2016*; *Yan and Bi, 2019*; *Tong et al., 2021*), which makes the tissue-scale flows that emerge from active elements coupled to the vertex model hard to predict using continuum approaches. Active nematic flows have been observed and modelled in in vitro epithelial tissues (*Saw et al., 2017*), and due to the locally contractile and extensile dynamics it is plausible that large-scale flows predicted by this model belong to the same class of models. In real tissues, however, the mechanisms that regulate how cells coordinate their internal mechanical stress directions in order to produce a stable flow pattern are unknown. Active vertex models therefore provide valuable new insights into the intricate interplay between mechanical and biochemical processes that control the collective cell behaviours in epithelial tissues.

We also briefly discuss how this work relates to other recent models for active junction contractions and convergence extension. The single-junction model shares a lot of common features with the model of *Dierkes et al., 2014*. The key difference, however, is a different myosin–tension feedback mechanism and that the dashpot in *Dierkes et al., 2014* was replaced with a Maxwell element. This was inspired by laser tweezer measurements of the response of cell–cell junctions in the *Drosophila* embryo to applied pulling force (*Clément et al., 2017*) which showed that the cellular junctions behave as a Maxwell viscoelastic material. The presence of an elastic spring attached to the dashpot introduces a viscoelastic timescale, which leads to the suppression of the oscillatory behaviour seen in *Dierkes et al., 2014*.

The model of *Staddon et al., 2019* considered a Maxwell element subject to active contraction, with the spring constant of the cellular junctions that constantly remodels itself to match strain in the junctions. The tension, however, remodels only if the strain exceeds a threshold value. These two features combine to provide a simple mechanism by which the junction can undergo ratchet-like behaviour and contract to a T1 event, and where T1 events can be triggered by applying external forces. Staddon et al., however, do not explicitly include kinetic equations for molecular motors.

Furthermore, the model of *Curran et al., 2017* introduced myosin-driven tension fluctuations in cell–cell junctions to cell ordering observed in experiments on the fly notum. They argued that myosin fluctuations combined with the isotropic distribution of myosin can either drive or inhibit T1 transitions. This suggests a robust mechanism by which epithelia can tune their properties. Similar conclusions about cell ordering have also been reached by *Krajnc et al., 2021* using a model with a mechanical feedback between junction contraction and force generation. Neither of those studies, however, explored whether myosin fluctuations could lead to directional deformations as observed during convergence–extension.

The two-dimensional models of *Wang et al., 2012* and *Lan et al., 2015* provide detailed descriptions for coupling between chemical signalling and corresponding mechanical responses. While they were able to produce T1 events, chemical anisotropy in the cell was externally imposed by tuning concentrations of relevant molecular species based on the origination of the junctions. This was also the case for the two-dimensional version of the model of *Staddon et al., 2019*, which was able to produce convergence–extension flows, albeit with time-dependent activity imposed in a given direction. Meanwhile, *Noll et al., 2017* have also introduced a vertex model with junctions that incorporate generic active feedback in a model for tissue contraction in *Drosophila* gastrulation. They did not, however, consider active T1 transitions.

Finally, we remark that pulling experiments on suspended cultured Madine Darby Canine Kidney (MDCK-II) cells (*Harris et al., 2012*) and on the *Drosophila* wing disk (*Duda et al., 2019*) do not show any topological transitions even for strains that exceed 50%. Instead, the deformation is accommodated by changes of cell shapes. In the case of MDCK layers, directional cell divisions also play an important role is releasing some of the mechanical stress introduced by the pull (*Wyatt et al., 2015*). While it is possible that T1 transitions would appear given sufficiently long time, it is unlikely that the mechanism proposed in this study would apply to these tissues. This is not unexpected since MDCK cells are transformed cells derived from adult tissues and likely not comparable to embryonic tissues that need to undergo large-scale highly coordinated shape changes involving massive cell rearrangements. The same argument holds for fly wing disk that also does not show large-scale anisotropic strain rates, driven by directed cell intercalations as are typical for embryonic tissues during gastrulation.

In summary, in this study we have introduced a mechanochemical model that describes the dynamics of active T1 transforms, that is, cell intercalation events that occur perpendicular to the

externally applied mechanical stress. Such processes are believed to play a key role in the primitive streak formation in avian embryos. Crucially, this study suggests that mechanical propagation of activation of myosin is a key ingredient in formation of the myosin tension chains that have been observed to accompany convergence–extension flow in chick embryo gastrulation (*Rozbicki et al., 2015*) and in *Drosophila* germ band extension (*Jacinto et al., 2002*). Results of this study show a good qualitative agreement with measurements on early-stage chick embryos.

We conclude by observing that it is remarkable that models that share the common assumption of a feedback between activity and mechanical tension are able to describe a range of markedly different biological processes in different organisms. Although the details depend on the specific biological system and its molecular details, this suggests that there is a set of universal physical mechanisms that govern tissue-scale behaviours.

## Materials and methods
### Model setup and analysis
#### Single active junction

To understand the mechanism that couples the kinetics of myosin motors to the local mechanical tension and leads to the activation of contractility in cell–cell junctions, we first analyse a model for a single junction. The surrounding tissue is abstracted by assuming that it provides an elastic, tension-generating background against which the junction actively contracts. There are two key ingredients that make an active junction. First, the junction is viscoelastic. Pull-release optical tweezer experiments on cell–cell junctions in *Drosophila* and chick embryos have shown that cell–cell junctions have a viscoelastic response (*Clément et al., 2017*; *Ferro et al., 2020*). This means that the junction is able to remove imposed tension by remodelling itself. Second, the junction can generate tension. Tension is generated by myosin motors that form mini filaments slide actin filaments past each other. The single-junction model provides insight into the conditions under which the junction length can contract to zero and trigger a T1 transition.

Specifically, the single junction (*Figure 1A*) is modelled as an active mechanochemical system comprising three components connected in parallel (*Figure 1—figure supplement 1A*): (1) a viscoelastic SLS element, (2) a viscous dashpot, and (3) a tension-sensitive force-generating motor. The junction is subject to external tension $T_{\text{ext}}$ produced and transmitted by the surrounding cells. The SLS element consists of an elastic spring of stiffness $B$ and rest length $a$ connected in parallel with a Maxwell element containing a spring of stiffness $k$ and rest length $l_0$ attached in series to a dashpot of viscosity $\eta$ (*Larson, 1999*). The spring $B$ captures the passive elastic response of the junction and models the effects of the surrounding tissue. In the two-dimensional model discussed below, this term arises naturally and accounts for changes in the cell area and perimeter. It is referred to as the *elastic barrier*. The Maxwell element models the viscoelastic nature of the junction and the ratio of viscosity and stiffness sets its relaxation timescale $\tau_{\text{v}} = \eta/k$. This is the timescale over which the junction remodels and adjusts its length to that imposed by the external load. The dashpot with viscosity $\zeta$ models dissipation with the environment. Finally, the junction is equipped with an active source of tension, which models the action of myosin motors.

Each molecular motor produces a force of magnitude $\tilde{\beta}$. $N_{\text{mot}}$ motors attached to actin filaments of the junction, therefore, generate tension $T_{\text{active}} = \tilde{\beta} N_{\text{mot}}$. If the maximum possible number of attached motors is $N_{\text{max}}$, $T_{\text{active}} = \beta m$, with $\beta = \tilde{\beta} N_{\text{max}}$ and $m = N_{\text{mot}}/N_{\text{max}}$. The junction is, however, a part of the tissue and in the absence of perturbations the average steady-state value of motors attached to all junctions $m_0 N_{\text{max}} \neq 0$. It is, therefore, appropriate to model the active tension as $T_{\text{active}} = \beta (m - m_0)$. This expression can also be understood as the leading-order term in the expansion of $T_{\text{active}}(m)$ around $m_0$ (*Dierkes et al., 2014*). For $m < m_0$, $T < 0$, that is, tension acts to extend the junction. While the $m_0$ term may appear counterintuitive, it reflects the fact that if motors attached to the junction are depleted, contractions of the surrounding junctions produce stronger pull on it than it can resist, resulting in elongation.

Tension feeds back on the kinetics of association and dissociation of myosin to actin filaments, leading to the kinetic equation for $m$,

$$\dot{m} = k_{\text{on}} - k_{\text{off}}(T) m, \tag{6}$$

where $k_{\text{on}}$ is the association rate constant, assumed to be tension-independent with no limit of the total available myosin, and $k_{\text{off}}(T)$ is the dissociation rate constant, assumed to be a monotonously decaying function of tension $T$. The kinetics of actin-bound myosin, therefore, resembles that of a catch bond. This assumption is motivated by measurements of binding and unbinding rates of myosin motors on single actin filaments and has been shown to be described as a simple negative exponential dependence of the dissociation rate on applied tension (*Veigel et al., 2003*; *Kovács et al., 2007*). The dynamics of the junction is given by the following set of equations,

$$\zeta\dot{l} = -T + T_{\text{ext}}, \quad \tau_{\text{v}}\dot{l}_0 = l - l_0, \quad \tau_{\text{m}}\dot{m} = 1 - mF(T), \tag{7}$$

with a natural choice being a sigmoid curve,

$$F(T) = \alpha + e^{-k_0(T-T^*)}, \tag{8}$$

where $\alpha > 0$ is the contribution to the myosin dissociation that does not depend on tension. The first equation describes the time evolution of the junction length due to the internal tension, $T = k(l - l_0) + B(l - a) + \beta(m - m_0)$, and external tension, $T_{\text{ext}}$. $T^*$ is the threshold tension, and $k_0$ controls the steepness of the $F(T)$ curve in the vicinity of $T^*$ (*Figure 1—figure supplement 1C*). The minus sign in front of the first term on the right-hand side indicates that $T > 0$ corresponds to a junction that is contracting, that is, $\dot{l} < 0$ for $T_{\text{ext}} = 0$. The second equation accounts for the viscoelastic nature of the junction (*Clément et al., 2017*), that is, the rest length $l_0$ relaxes towards the actual length $l$ with a characteristic timescale $\tau_{\text{v}}$. Finally, the third equation was obtained by dividing *Equation 6* by $k_{\text{on}}$, where $\tau_{\text{m}} = 1/k_{\text{on}}$ is the timescale of myosin association. In general, binding and unbinding of molecular motors is a stochastic process and the third equation should also include stochastic terms. For simplicity, such terms were omitted here, but were included in the two-dimensional model. Furthermore, it is assumed that $l$ and $l_0$ are comparable in magnitude, that is, that $(l - l_0)/l_0 < 1$ which makes using a linear spring model appropriate despite the total length of the junction changing significantly as the junction collapses.

Finally, in all simulations of the single-junction model $k$, $\zeta$, and $a$ were kept fixed and, therefore, length is measured in units of $a$, time in units of $t^* = \zeta/k$, and force in units of $ka$. Parameters and their values used in the analysis of the single junction are listed in *Table 1*.

## Vertex model with active junctions

The mechanical response of the tissue is modelled with the vertex model (*Farhadifar et al., 2007*; *Fletcher et al., 2014*). The associated mechanical energy is a function of the cell area and perimeter,

$$E_{\text{VM}} = \sum_{\text{C}} \left[ \frac{\kappa_{\text{C}}}{2}(A_{\text{C}} - A_0)^2 + \frac{\Gamma_{\text{C}}}{2}(P_{\text{C}} - P_0)^2 \right], \tag{9}$$

where $A_C$ and $P_C$ are the area and the perimeter of cell $C$, respectively, $A_0$ and $P_0$ are the preferred area and perimeter, respectively (assumed to be the same for all cells), and the sum is over all cells. The first term in *Equation 9* accounts for three-dimensional incompressibility of cells, and $\kappa_{\text{C}}$ is the corresponding elastic modulus of the cell $C$. The second term in *Equation 9* contains a combination of actomyosin contractility in the cell cortex and intercellular adhesions, where $\Gamma_{\text{C}}$ is the contractility modulus of cell $C$ (*Farhadifar et al., 2007*).

All inertial effects were neglected, and, in line with the existing literature, it is assumed that the friction can be modelled as viscous drag on each vertex. The equation of motion for vertex $i$ is, therefore, a balance between friction and mechanical forces,

$$\zeta\dot{\mathbf{r}}_i = -\nabla_{\mathbf{r}_i}E_{\text{VM}} + \mathbf{F}_{\text{active}}, \tag{10}$$

where $\zeta$ is the friction coefficient, and $\mathbf{F}_{\text{active}}$ accounts for all active forces. Stochastic forces are, however, omitted since those do not qualitatively affect the dynamics at timescales of interest. Inserting *Equation 9* into *Equation 11* leads to

$$\dot{\mathbf{r}}_i = \frac{1}{\zeta}\sum_{\text{e}}\left(F_{\text{e}}^{\text{A}}\mathbf{e}_z \times \mathbf{l}_{\text{e}} + T_{\text{e}}^{\text{P}}\hat{\mathbf{l}}_e\right) + \frac{1}{\zeta}\mathbf{F}_{\text{active}}, \tag{11}$$

with $F_e^A = \frac{1}{2}\left(p_{C_{e,l}} - p_{C_{e,r}}\right)$ and $T_e^P = -\left(t_{C_{e,l}} + t_{C_{e,r}}\right)$ being the magnitudes of, respectively, the area and perimeter contributions to the force due to junction $e$. The subscript $C_{e,l}$ ($C_{e,r}$) denotes the cell to the left (right) of the junction $e$ when facing in the direction of $\mathbf{l}_e$. $\mathbf{e}_z$ is the unit-length vector perpendicular to the plane of the tissue and the vector $\mathbf{l}_e$ points along the junction $e$ away from vertex $i$, $\hat{\mathbf{l}}_e = \mathbf{l}_e/l_e$ with $l_e = |\mathbf{l}_e|$, $p_C = -\partial E_{VM}/\partial A_C = -\kappa_C\left(A_C - A_0\right)$ is the hydrostatic pressure on the cell C, and $t_C = -\partial E_{VM}/\partial P_C = -\Gamma_C\left(P_C - P_0\right)$. Finally, the sum is over all junctions that originate at vertex $i$ and terms in the sum appear in counterclockwise order (*Figure 2—figure supplement 1A*).

Activity is introduced by assuming that each junction contains two active elements supplied by the two cells sharing it. Furthermore, each cell is assumed to have a finite pool of myosin, $M$. The finite pool of myosin acts to introduce correlations in the distribution of junctional myosin within each cell. In other words, it models the mechanism by which, for example, depletion of myosin on a given junction is correlated to myosin accumulation on its neighbouring junctions. This is of central importance to establish myosin anisotropy within the cell. There are clearly many alternative ways to model coupling of myosin on different junctions within a cell. The model used here, however, requires a minimal number of parameters while being biologically plausible.

Of this total myosin, $M$, a fraction $m_{act}^C = \sum_{e=1}^{z_C} m_e^C$ is assumed to be activated, that is, bound the junctions, and thus depleted from the pool. Here, $z_C$ is the number of junctions shared by the cell $C$. The association rate of myosin to an individual junction is proportional to $M - m_{act}^C$. As in the single-junction model, the dissociation rate is proportional to the amount of myosin bound to the junction $e$, modulated by a tension-dependent function $F\left(T_e\right)$. To match the steady-state value of myosin of the single junction where $m_{eq} = F\left(T\right)^{-1}$, the prefactor of the unbinding term also needs to be $z_C$. Then, to lowest order in the contributions of the coupled junctions, $m_{eq} = \left(2F\left(T\right)\right)^{-1}$. We can recover $m_{eq} \approx 0.5$ at $T = T^*$ by choosing small but finite values of $\alpha$ ($\alpha = 0.1$ in simulations). This leads to the kinetic equation for the myosin motor attached to the junction $e$ by cell $C$,

$$\tau_m \dot{m}_e^C = \left(M - m_{act}^C\right) - z m_e^C F\left(T_e\right) + \eta_e^C, \tag{12}$$

where it is assumed that $M$ and $z$ are same for all cells. Without loss of generality, it is possible to set $M = z$. $\tau_m$ is the inverse rate of myosin binding, that is, the timescale of attachment of myosin motors and $\eta_e^C$ is a random white noise with zero mean and variance

$$\langle \eta_e^C\left(t\right) \eta_e^C\left(t'\right)\rangle = f\delta\left(t - t'\right), \tag{13}$$

which accounts for the stochastic nature of myosin binding and unbinding. The noise term is important for the system to be able to break the symmetry imposed by using regular hexagonal tilings. It is, however, not strictly necessary to introduce noise to the myosin kinetics, but instead consider, for example, that mechanical properties of the cells are randomly distributed. While the quantitative results would be affected, such a model is not expected to have qualitatively different behaviour compared to what is discussed here.

*Equation 9* models the passive elastic response of the tissue. In order to achieve an active T1 event, remodelling needs to be present, that is, the system must be viscoelastic. There are various ways to include viscoelastic effects into the vertex model. In order to be consistent with the single-junction model, it is further assumed that the junction $e$ has a viscoelastic contribution to the tension, $k_e\left(l_e - l_e^0\right)$, where $k_e$ is the spring constant analogue to the elastic part of the Maxwell element in the single-junction model and $l_0$ is the time-dependent rest length with dynamics,

$$\tau_v \dot{l}_e^0 = l_e - l_e^0, \tag{14}$$

where $\tau_v$ is the characteristic timescale for viscoelastic remodelling. The full expression for the tension of junction $e$ is

$$T_e = T_e^P + k_e\left(l_e - l_e^0\right) + \beta_e^{C_l}\left(m_e^{C_l} - m_0\right) + \beta_e^{C_r}\left(m_e^{C_r} - m_0\right). \tag{15}$$

As above, the superscript $C_l$ ($C_r$) denotes the cell to the left (right) of the junction $e$ when facing in the direction of $\mathbf{l}_e$. $\beta_e^C$ is the activity of the junction $e$ produced by the cell $C$, that is, it is a constant with units of force that measures the strength of the mechanochemical coupling and $m_0$ has the same

meaning as in the single-junction model. For simplicity, $\Gamma_C \equiv \Gamma$ for all cells and $k_e \equiv k$ for all junctions and both parameters were kept constant in all simulations. The unit of length, $a$, was chosen to be the length of the side of a regular hexagon, which allows us to set the unit of time $t^* = \zeta/(\Gamma + k)$, and the unit of force $f^* = (\Gamma + k)a$. *Equation 11*, with $T_e^P$ given by *Equation 15*, and *Equation 12* and *Equation 14* describe the dynamics of the vertex model with active junctions.

The equations of motion for the two-dimensional model were integrated numerically for a rectangular patch made of $N = 158$ hexagonal cells and patch made of $N = 600$ randomly shaped cells using open boundary conditions. Mechanical anisotropy was created by applying a force $\mathbf{f}_{pull} = \pm f_{pull}\mathbf{e}_x$ to the left and right boundary vertices (*Figure 2A*), where the positive (negative) sign corresponds to the right (left) boundary. The initial pull was applied for $10^3 t^*$ with both activity and viscoelastic relaxation switched off, sufficient to reach mechanical equilibrium. Once the system reached an equilibrium stretched state, activity was switched on in 14 central cells in the case of the hexagonal patch and for 520 cells in the case of the random tissue patch. For the hexagonal case, activity was set to $\beta$ in 4 cells and to $\beta/2$ in 10 'buffer' cells surrounding them, in order to suppress numerical instabilities at contacts between active and passive cells. For the random patch, activity was set to $\beta$ in all cells except for a single-cell thick layer of boundary cells that were kept passive to prevent artefacts due to tension chains reaching the sample boundary. An external pulling force of constant magnitude was applied throughout the entire simulation. The active system was simulated for $\max\left(1600t^*, 10\tau_m, 10\tau_v\right)$ using time step $10^{-2}t^*$. In the hexagonal case, the dynamics of the central horizontal junction and the shoulder junctions shared by the four central active cells was monitored. The orientation of the hexagonal lattice was chosen such that the central active junction was parallel to the direction of the applied external force. In the case of the random patch, the dynamics of all junctions shared by active cells was monitored.

Parameters and their values used in the analysis of the vertex model with active junctions are listed in *Table 2*.

## Characterisation of T1 transitions and tissue flow

The T1 events were implemented following the procedure proposed by *Spencer et al., 2017*, where a junction shorter than $0.02a$ collapses into a fourfold vertex. The fourfold vertex either remains stable or it is resolved into two threefold vertices based on the sum of the forces acting along the four junctions connected to it. Importantly, the direction of the new junction is not imposed, and this procedure does not generally lead to a new junction orthogonal to the collapsed one. For simplicity, vertices with connectivity greater than four were not considered.

If a T1 transition occurs, it is necessary to assign myosin to the newly created junction and redistribute the myosin associated to the collapsed junction to the surrounding junctions. As illustrated in and using the notation introduced in *Figure 2—figure supplement 1B*, the myosin $m_e$ of the collapsing junction is stored right before the junction collapses. After the T1 transition, the myosin on the new junction was set to $m_0$. Both inner shoulder junctions increased their myosin by $m_e/2$, while the myosin on the outer junctions was reduced by $\min(m_2, m_0/2)$, where $m_2$ is its myosin right before the T1 transition and the minimum function ensures that myosin remains $\geq 0$. The myosin redistribution procedure reduces artificial jumps of the junctional myosin as the system progresses through T1 transition and also provides a natural way to handle the finite myosin pool. After the transition, $l_0$ of the new junction is set to $0.022a$.

To identify and characterise T1 transitions, and quantify the associated deformation of the model tissue, we used the analysis method introduced by *Graner et al., 2008*. For cellular patterns, three tensors are defined, texture ($\hat{M}$), geometrical texture change ($\hat{B}$), and topological texture change ($\hat{T}$). Tensor $\hat{M}$ describes the shape of the current cell configuration, and it is defined as

$$\hat{M} = \langle \hat{m} \rangle = \langle \boldsymbol{\ell} \otimes \boldsymbol{\ell} \rangle = \begin{pmatrix} \langle X^2 \rangle & \langle XY \rangle \\ \langle YX \rangle & \langle Y^2 \rangle \end{pmatrix}, \tag{16}$$

where $X$ ($Y$) is the $x$ ($y$) component of the vector $\boldsymbol{\ell} = \mathbf{r}_2 - \mathbf{r}_1$ connecting centroids of two neighbouring cells at positions $\mathbf{r}_1$ and $\mathbf{r}_2$, respectively, $\langle \cdot \rangle = \frac{1}{N_{tot}}\sum(\cdot)$ is an average over $N_{tot}$ pairs of neighbours, that is, cell–cell contacts, and $\hat{m} = \boldsymbol{\ell} \otimes \boldsymbol{\ell}$. Using the same notation, the tensor $\hat{B}$ describes shape changes of the cell configuration during a time interval $\Delta t$, and it is defined as

$$\hat{B} = \hat{C} + \hat{C}^{\mathrm{T}}, \tag{17}$$

where the superscript T denotes the matrix transpose and

$$\hat{C} = \left\langle \ell \otimes \frac{\Delta \ell}{\Delta t} \right\rangle = \begin{pmatrix} \left\langle X \frac{\Delta X}{\Delta t} \right\rangle & \left\langle Y \frac{\Delta X}{\Delta t} \right\rangle \\ \left\langle X \frac{\Delta Y}{\Delta t} \right\rangle & \left\langle Y \frac{\Delta Y}{\Delta t} \right\rangle \end{pmatrix}. \tag{18}$$

Finally, the tensor $\hat{T}$ identifies T1 transitions by quantifying topological changes of the cell configuration in the time interval $\Delta t$ via tracking appearance and disappearance of contacts between cells. It is defined as

$$\hat{T} = \frac{1}{\Delta t} \langle \hat{m} \rangle_{\mathrm{a}} - \frac{1}{\Delta t} \langle \hat{m} \rangle_{\mathrm{d}}, \tag{19}$$

where $\langle \cdot \rangle_{\mathrm{a}}$ ($\langle \cdot \rangle_{\mathrm{d}}$) is the average over contacts that appeared (disappeared) during the time interval $\Delta t$. The total number of contacts that appear and disappear is typically much smaller than $N_{\mathrm{tot}}$, which means that $\hat{T}$ data can be quite noisy. While all three tensors can be calculated for individual cells, the averaging is meant to be carried over a mesoscopic region. We, therefore, averaged over the 14 central active cells (*Figure 2A*, dark-shaded cells), as well as over an ensemble of $n = 32$ noise realisations. For the disordered tissue, we averaged over all $N = 520$ active cells.

Tensors $\hat{M}$, $\hat{B}$, and $\hat{T}$ all involve averaging over cell–cell contacts and, therefore, describe a discrete system. In order to make connections to continuous deformations of the entire tissue, one introduces their continuous counterparts, the statistical strain tensor ($\hat{U}$), the velocity gradient tensor ($\hat{V}$), and the tensor of the rate plastic deformations (i.e. topological rearrangements rate) ($\hat{P}$) (*Graner et al., 2008*). These tensors are defined as

$$\hat{U} = \frac{1}{2} \left( \log \hat{M} - \log \hat{M}_0 \right), \tag{20}$$

where $\hat{M}_0$ is the texture tensor of an arbitrary reference configuration for the statistical relative strain – we chose the initial undeformed configuration. Further,

$$\hat{V} = \frac{1}{2} \left( \hat{M}^{-1} \hat{C} + \hat{C}^{\mathrm{T}} \hat{M}^{-1} \right), \tag{21}$$

and

$$\hat{P} = \frac{1}{2} \left( \hat{M}^{-1} \hat{T} + \hat{T} \hat{M}^{-1} \right). \tag{22}$$

In the case when variations in $N_{\mathrm{tot}}$ can be neglected, one can show that (*Graner et al., 2008*)

$$\hat{V} = \frac{\mathcal{D} \hat{U}}{\mathcal{D} t} + \hat{P}, \tag{23}$$

where $\mathcal{D}/\mathcal{D}t$ is the corotational derivative (*Larson, 1999*). This equation just states that the velocity gradient is a sum of two contributions, reversible changes of the internal strain and the rate of irreversible plastic rearrangements. The significance of *Equation 23* is that if one assumes that there are no plastic events other than T1 transitions, it is possible to obtain the total strain of the system $\hat{\varepsilon}^{\mathrm{tot}}$ by integrating the tensor $\hat{V}$ over time, that is,

$$\hat{\varepsilon}^{\mathrm{tot}} = \int_{t_0}^{t_{\mathrm{tot}}} \mathrm{d}t' \hat{V}(t'), \tag{24}$$

where $t_0$ is the time when the activity is switched on and $t_{\mathrm{tot}}$ is the total simulation time. The difference $\varepsilon_{\mathrm{xx}}^{\mathrm{tot}} - \varepsilon_{\mathrm{yy}}^{\mathrm{tot}}$ of the $xx$ and $yy$ components of $\hat{\varepsilon}^{\mathrm{tot}}$ (i.e. total strains in the $x$ and $y$ direction, respectively) was used as the measure of the amount of convergence–extension in the tissue induced by the active T1 transition. *Figure 3—figure supplement 1* shows ensemble-averaged time traces of

$\hat{U}$ and the time-integrated total strain $\hat{V}$ and plastic strain $\hat{P}$ through the active T1 transition and the subsequent propagation phase.

## Convergence–extension in a patch of randomly shaped active cells

We used the two-dimensional model with parameters for the cell mechanics and the myosin feedback loop as defined in *Table 2* on a tissue patch with cells of random shapes. The patch was generated using an iterative procedure. First, $N = 600$ points were placed at random in a square region of size $L$ chosen such that the average cell area is equal to that of the hexagonal patch, that is, $A_0 = 2.598a^2$. During the initialisation process, we ensured that no two points are closer than $a$ to each other. Once all points are placed in the box, the Voronoi diagram was constructed and centroids of each cell of the Voronoi diagrams were used as the seeds for constructing Voronoi diagram in the next iteration. This was repeated until no centroid moved more than $5 \times 10^{-5}a$ between two consecutive iterations, resulting in a centroidal Voronoi tessellation. Centroidal Voronoi tessellations have several convenient properties. For example, typically, there are no outliers (i.e. very large or very small cells are unlikely) and the distribution of cell neighbours is remarkably similar to actual epithelia. Finally, to avoid the spontaneous formation of an actomyosin cable at the outside border, we used a setup where the $N = 520$ interior cells are active, and a one-cell-thick layer of passive cells forms the boundary.

We used the same stretching protocol as for the ordered patch. At every vertex on the left and right boundaries, the force $f_{\text{pull}}$ was exerted during the entire duration of the simulation. The passive tissue was first made anisotropic by stretching it for $10^3 t^*$ with activity and the viscoelastic relaxation turned off. We then turned on activity and viscoelasticity and simulated the system for $1.6 \times 10^3 t^*$. This protocol was repeated for a range of activities $\beta$, viscous relaxation times $\tau_v$, and myosin times $\tau_m$ (not shown since like we observed for the single active T1, results for convergence–extension were largely independent of $\tau_v$ and $\tau_m$ and quantitatively similar to *Figure 7*). All results were averaged over $n = 5 - 33$ different random initial configurations and with different random number generator seeds for the myosin noise. The results shown in *Figure 7C and D* are reported with a 95% confidence interval computed using bootstrapping by resampling $10^3$ times.

*Figure 7* shows tissue dynamics for $\beta = 0.5f^*$ and $f_{\text{pull}} = 0.2f^*$, which is the optimal region for convergence–extension. We measured convergence–extension using the integral of $\hat{V}$, where the calculation was done over the entire active region, and the starting point was the time when the activity was turned on. For the elastic strain $\hat{U}$, we used the undeformed disordered initial condition to define the reference strain $\hat{M}_0$. *Figure 7—figure supplement 1* shows the time traces of the ensemble averaged tensors for elastic strain $\hat{U}$, integrated total strain $\hat{V}$, and integrated plastic strain $\hat{P}$.

We also quantified the amount of tension and myosin anisotropy using the tissue-averaged stress tensor components

$$\hat{\mathbf{T}}_{\text{tension}} = \left\langle \frac{1}{A_C} \sum_{e \in C} T_e \hat{\mathbf{l}}_e \times \mathbf{l}_e \right\rangle \qquad (25)$$

and

$$\hat{\mathbf{M}}_{\text{myo}} = \left\langle \frac{1}{A_C} \sum_{e \in C} m_e \hat{\mathbf{l}}_e \times \mathbf{l}_e \right\rangle. \qquad (26)$$

To measure convergence–extension strain and anisotropy of shape, myosin, and tension, we chose $t = 700t^*$, which is the time point where convergence–extension and anisotropy stabilise, giving the results shown in *Figure 7C and D*.

To measure orientations of T1s, we first identified cells involved in a T1 transition by diagonalising the $\hat{\mathbf{T}}$ tensor associated with active cells at a given instance in time. The signature of a T1 event is a non-zero $\hat{\mathbf{T}}$, and the sign of its trace determines if a junction appeared or disappeared. The eigenvector corresponding to the largest eigenvalue determines the direction of the event. We tracked all of such events over the simulation runtime and generated polar histograms such as the one shown in *Figure 7B*. As shown in *Figure 7—figure supplement 2A* for $\beta = 0.5f^*$ and $f_{pull} = 0.2f^*$, there are temporally strongly correlated back-and-forth T1 transitions at the same angle. These correspond to four cells flipping back and forth through a T1 transition, or 'flickering' in the videos. As these events are artefacts of the simulation, we excluded them from the data. *Figure 7—figure supplement 2B*,

which was averaged over $n = 5$ independent simulations, shows that flickering events have mostly been filtered out of the dataset.

*Figure 7—figure supplement 2* shows filtered T1 histograms for other mechanical conditions, outside the parameter region of where convergence–extension occurs, averaged over $n = 5$ independent simulations. In particular, we also include a passive tissue that flows in the direction of the pulling, and an active isotropic tissue without applied forces where the T1 distribution is isotropic. This last situation strongly resembles the observed T1 distribution in the anterior region of the streak (*Figure 7—figure supplement 3C*). As shown in *Figure 8—figure supplement 1*, this is also a mostly isotropic tissue.

## Experimental data analysis

### Experimental data

Active myosin (phosphorylated myosin light chain) and actin staining in fixed embryos was performed as described in *Rozbicki et al., 2015*. Embryos of a transgenic chick line with cell membranes of all cells in the embryonic and extra embryonic tissues labelled with a green fluorescent protein tag (myr-EGFP) were live-imaged using a dedicated light-sheet microscope as described previously (*Rozbicki et al., 2015*). The microscope produces cell-resolution images of the entire embryo with time resolution of 3 min between frames for periods up to 16 hr (stage EGXIII-HH4). $n = 6$ rectangular areas of interest of size $\approx 255 \times 220 \ \mu m^2$ were chosen to lie next to each other along the sickle-shaped mesendoderm precursor region in the embryo's posterior, perpendicular to the direction of the forming primitive streak. The two central sections were chosen to lie in the middle of the sickle region that initiates the formation of the streak. Each area of interest was tracked for $\approx 4.6$ hr covering the onset of the flows driving streak formation. The average motion of the area of interest, determined via particle image velocimetry using PIVlab (*Thielicke and Sonntag, 2021*), was used to track its displacement to be able to follow the same patch of cells over time. After this, these image time series were bandpass filtered to remove some noise followed by segmentation using the watershed algorithm in MATLAB. To follow the same cells over time, their centroid positions calculated form the segmentation of the first image of the time series are projected forward to the next frame using a newly calculated high-resolution velocity field between these successive images and using these as seed points for the segmentation of the next image (*Rozbicki et al., 2015*). This procedure allowed us to track individual cells between consecutive time frames and determine changes in neighbours over time as well as determine the directions of appearing and disappearing junctions associated with T1 transitions. The $\hat{\mathbf{M}}$, $\hat{\mathbf{U}}$, and $\hat{\mathbf{V}}$ tensors in a particular area of interest were averaged for all cells in centred circular domains of 190 μm diameter between successive pairs of segmented images and averaged over 30 min time intervals.

The spatially averaged $\hat{\mathbf{M}}(t)$ texture tensor allowed us to both compute $\hat{\mathbf{U}}$ and also to directly quantify shape anisotropy from the eigenvalues $(m_L, m_S)$ and eigenvectors $\left(\xi_L^M, \xi_S^M\right)$ of the time-averaged $\langle \hat{\mathbf{M}}(t) \rangle_t$, where $L$ and $S$ label the large and small components, respectively. We defined the dimensionless shape anisotropy shown in *Figure 8C* in the main text as

$$p_S = \frac{m_L - m_S}{m_L + m_S},$$

(27)

and the shape anisotropy direction as the angle $\xi_L^M$ makes with the lab frame $x$-axis.

Unlike in the simulation, in the real embryo, cells divide, ingress, and flow in and out of the region of interest. Therefore, it is not immediately clear what to use as the reference texture tensor $\hat{\mathbf{M}}_0$ when computing $\hat{\mathbf{U}}$ tensor. For simplicity, we chose the isotropic tensor constructed from the time-averaged eigenvalues as

$$\hat{\mathbf{M}}_0 = \frac{1}{2}(m_S + m_L)\hat{\mathbf{I}}.$$

(28)

To compute the flow anisotropy, we measured the integrated total strain tensor

$$\hat{\varepsilon}^{\text{tot}}(t) = \int_{t_0}^{t} dt' \hat{V}\left(t'\right),$$

(29)

analogous to the simulated tissue patches. Similar to $\hat{\mathbf{M}}$, we can then compute the eigenvalues $(V_L, V_S)$ and eigenvectors $(\xi_L^V, \xi_S^V)$ of the final $\hat{\varepsilon}^{\text{tot}}(t_{max})$, where we again label the large and small components $L$ and $S$, respectively. Here, we now typically have $V_S < 0$ and $V_L > 0$, corresponding to the convergence and extension directions of the tissue, respectively. We compute the total flow magnitude as

$$\varepsilon_{C-E}^{\text{tot}} = V_L - V_S, \tag{30}$$

and the flow anisotropy direction as the angle $\xi_L^V$ makes with the lab frame *x*-axis (*Figure 8C*).

For comparison, in *Figure 8—figure supplement 1*, we show the integrated $\hat{V}$ and the $\hat{U}$ tensor for a region in the anterior of the embryo which does not undergo convergence–extension.

The phosphomyosin light chain staining patterns shown in *Figure 8A* are representative of the patterns observed in over 100 embryos of similar developmental stages. The intercalation data shown in *Figure 8C* are based on analysis of a light-sheet microscopy sequence of a single embryo, representative of over 50 experiments of control embryos.

Fertile eggs (Shaver brown) were obtained from Henri Stewart & Co Ltd, UK. Fertile Myr-GFP eggs were obtained from the National Avian Research Facility at the Roslin Institute, University of Edinburgh, UK.

## Acknowledgements

RS, MC, and CJW acknowledge support by the UK BBSRC (award BB/N009789/1). SH and IDC acknowledge support by the UK BBSRC (grant number BB/N009150/1-2). IDC acknowledges funding under Dioscuri, a programme initiated by the Max Planck Society, jointly managed with the National Science Centre in Poland, and mutually funded by the Polish Ministry of Science and Higher Education and German Federal Ministry of Education and Research (UMO-2019/02/H/NZ6/00003). We thank Antti Karjalainen for providing the MATLAB code used to analyse experimental data. RS thanks Andrej Košmrlj, Daniel Matoz-Fernandez, and Sijie Tong for many helpful discussions about the vertex model.

## Additional information

### Funding

| Funder | Grant reference number | Author |
| --- | --- | --- |
| Biotechnology and Biological Sciences Research Council | BB/N009789/1 | Rastko Sknepnek Manli Chuai Cornelis Weijer |
| Biotechnology and Biological Sciences Research Council | BB/N009150/1-2 | Ilyas Djafer-Cherif Silke Henkes |

The funders had no role in study design, data collection and interpretation, or the decision to submit the work for publication.

### Author contributions

Rastko Sknepnek, Conceptualization, Software, Formal analysis, Supervision, Funding acquisition, Validation, Investigation, Visualization, Methodology, Writing - original draft, Project administration; Ilyas Djafer-Cherif, Data curation, Formal analysis, Investigation, Visualization, Writing – review and editing; Manli Chuai, Formal analysis, Investigation, Methodology, Writing – review and editing; Cornelis Weijer, Conceptualization, Resources, Data curation, Formal analysis, Supervision, Funding acquisition, Validation, Investigation, Methodology, Project administration, Writing – review and editing; Silke Henkes, Conceptualization, Formal analysis, Supervision, Funding acquisition, Validation, Investigation, Visualization, Methodology, Writing - original draft, Project administration

### Author ORCIDs

Rastko Sknepnek http://orcid.org/0000-0002-0144-9921

Ilyas Djafer-Cherif · http://orcid.org/0000-0002-9619-0202
Cornelis Weijer · http://orcid.org/0000-0003-2192-8150
Silke Henkes · http://orcid.org/0000-0002-6688-7367

## Decision letter and Author response
Decision letter https://doi.org/10.7554/eLife.79862.sa1
Author response https://doi.org/10.7554/eLife.79862.sa2

## Additional files

### Supplementary files
• MDAR checklist

### Data availability
The current manuscript is primarily a computational study, so no data have been generated for this manuscript. Modelling code is publically (GNU public license v 2.0) available on GitHub at: https://github.com/sknepneklab/ActiveJunctionModel (copy archived at *Sknepnek, 2023*). The experimental data presented in Figure 8 and Figure 8—figure supplement 1 has been generated as described in the methods section.

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
