## [Editor Report]

This theoretical investigation provides important findings on the role of active mechanical feedback on tissue remodelling. The authors present convincing evidence that mechanically enforced myosin recruitment at cell-cell junctions can lead to tissue expansion in the direction perpendicular to an externally applied uniaxial mechanical stress. The relevance of the proposed mechanism for convergence–extension systems requires more investigation through comparison with experimental data.

---

## [Decision Letter]

**Decision letter after peer review:**

Thank you for submitting your article "Generating active T1 transitions through mechanochemical feedback" for consideration by *eLife*. Your article has been reviewed by 3 peer reviewers, one of whom is a member of our Board of Reviewing Editors, and the evaluation has been overseen by Aleksandra Walczak as the Senior Editor. The reviewers have opted to remain anonymous.

Essential revisions:

1) Improve the comparison with experiments (see reports).

2) Argue more convincingly for biological relevance.

3) Put the positive feedback tension/myosin more into context and discuss possible effects of mechanisms that are an alternative to the one you implemented (see reports).

*Reviewer #1 (Recommendations for the authors):*

This paper presents a numerical study of tissue convergent extension during epithelial morphogenesis. It builds upon several earlier works, including so-called vertex models, and single junction mechanical models. The specificity of the model presented lies in its active component: a positive feedback exists between Myosin activity and mechanical tension in each junction. This leads to interesting tissue-scale dynamics. Upon tissue pulling, the authors report conditions for which intercalation occurs perpendicular to the pulling direction. They provide a rather detailed analysis of the model and its numerical outcome, and a qualitative comparison with convergent extension in avian embryos.

Strengths:

The model is presented in a rigorous manner and with a lot of details and pedagogy. It is discussed in comparison with other models of the field, and the similarities and differences are clearly and honestly presented by the authors in their discussion.

The numerical study provides interesting and possibly counter-intuitive predictions in which tissue convergent extension can occur perpendicular to an external pulling axis, due to a positive feedback between myosin motors recruitment and tension which leads to the formation of Myosin chains (or cables) along the pulling axis.

The conditions in which this regime of tissue remodeling can occur (under the hypotheses of the model) are thoroughly investigated. Finally the authors go a bit beyond the sole numerical study, and analyze experimental data of the polarized tissue flow occurring upon primitive streak formation in avians.

Weaknesses:

Although the numerical investigation is thorough and the model hypotheses are clearly stated, the paper falls a bit short of convincing readers that the emergent behaviors observed in the model are of actual biological relevance.

The hypotheses and ingredients of the model are clearly stated (1) junctions are viscoelastic, (2) junctions are active, i.e. they can generate tension through myosin, and (3) positive feedback between tension and myosin. Yet the 3rd hypothesis, which gives its singularity to the model, is not discussed enough, and its relevance to well-known convergent extension systems is not analyzed. In the paper, the positive feedback arises from the decrease of unbinding rate upon tension increase. This "catch bond" hypothesis (note: the term catch-bond only appears once, in the methods) is at the core of all the results obtained in the paper, and this should be clearly stated and discussed. As for the positive feedback, one could also imagine that it is not the unbinding rate that decreases upon tension increase, but rather the myosin recruitment rate that increases. Would that be formally equivalent? The authors should address this point and its biological implications.

The comparison between the model outcome (notably, the T1 transitions preferably oriented in the direction perpendicular to pulling) and existing experimental data is a bit elusive. The authors propose a semi-quantitative comparison with convergent-extension occurring upon primitive streak formation in avians. The data and conclusion is not fully convincing, partly because of the polarization data itself (Figure 8C), and partly because it's not clear what would be the equivalent of the external pulling force in this self-polarizing flow.

The authors also mention the germband of *Drosophila*, but the model is, I believe, not relevant for that system. Although there is indeed an external force pulling the tissue (pulling posterior midgut), polarization is perpendicular and extension parallel to the pulling force (the opposite of the model outcome).

In that spirit, they should also discuss pulling experiments (fly wing disc / Mao lab, suspended epithelia / Charras lab) where it's not so clear that topological transitions occur upon pulling.

Finally, the paper builds upon already existing models of junction and tissue mechanics. Beyond the vertex model itself, the hypothesis of a feedback between myosin activity and tension has also been around in the field for some time, including in modeling papers (which authors acknowledge honestly).

In conclusion:

The paper presents a thorough numerical analysis of T1s and convergent extension upon tissue pulling when a positive feedback exists between Myosin and junction tension. Yet, the fact that this type of model and hypotheses are not completely new, combined to a lack of comparison to experimental data (except for avian gastrulation) makes it a bit difficult for the reader to assess the novelty of the findings presented in the paper.

I liked the presentation of the model a lot, the details given, the pedagogy, the comparison with what already existed (Discussion section), and the thorough analysis of the model outcome.

On the other hand, I find it a bit difficult to see whether the novelty is strong enough for publication in *eLife*:

First, because somewhat similar models exist in the field. According to the authors, the specificity here is the inclusion of the positive feedback for the active part of the model. Yet, other models that the authors cite include a similar type of feedback, although they might not specifically focus on topological transitions and convergent extension. In addition, the existence of such a feedback (and its possible consequences on tissue morphogenesis) has been around in the field for a while, under various forms (here, the authors chose a catch-bond type dissociation rate for myosin).

Second, because the comparison (even qualitative) with existing experimental data remains elusive.

Hence the major finding of this paper is that a tissue in which a positive feedback between tension and motor recruitment exists can display active intercalations and convergent extension perpendicular to an external pulling force. In which experimental systems this prediction could be relevant remains to be determined.

– Does the positive feedback tension/myosin need to stem from the catch bond model? Could one imagine increased motor recruitment open junction/cell stretch (This would be in line with some experimental data, including stretching experiments). Would that be formally equivalent to the catch bond, or would that change the outcome of the model? If yes, how?

– In which systems do people observe tissue extension along the direction perpendicular to pulling? The authors only refer to the avian embryo in which the nature of the flow (embryo-scale) makes it a bit difficult (at least for me) to interpret what is the chicken and what is the egg. A lot of data is now available out there for analysis. It would be great to see the same kind of analyses in other experimental systems. Notably, the fly germband (that the authors rapidly mention) seems to behave completely differently, with cables of Myosin perpendicular to the pulling axis (pulling force being exerted by the posterior midgut). How about convergent extension in the zebrafish or *Xenopus* embryos?

– What happens at the boundaries? Here we look at a central patch of cells. And somewhat related question, how would that work in a closed geometry (such as the fly embryo). I feel that the avian discussion (paragraph 1 of discussion) could be expanded with considerations on how the feedback can produce long range flows even though only a small patch of cells is initially "activated".

– Authors should cite Duda et al. (Dev Cell 2019) from the Mao group. In this article it is shown that Myosin polarizes and form cables upon external tissue stretch in the fly wing disc. This is fully relevant to their model and it is somewhat surprising that they do not cite it. Disclaimer: I'm not on that paper

– The authors present their model as "one of the simplest descriptions" that features the three main ingredients required (viscoelastic junctions, active junctions, positive feedback myosin/tension). What would be the simplest and why didn't they choose it?

– Authors should better explain the rationale of the "elastic barrier" term B(l-a) , as compared to the k(l-l0) term.

*Reviewer #2 (Recommendations for the authors):*

Sknepnek et al. study a model for junction dynamics in a two-dimensional vertex model, intended to describe epithelial mechanics. In the model, a concentration of myosin on each half-junction evolves according to a chemical balance equation and induces an active tension in the junction. The chemical balance equation incorporates a mechanical feedback, whereby myosin unbinding is inhibited at high tension. The authors first study a single junction and consider thresholds for the junction to fully contract. They then investigate the behaviour of a few « active » junctions in a passive network and a fully active vertex model. They find that at intermediate external pulling force and myosin-induced tension, the simulated epithelium can undergo convergence-extension against the pulling force.

The manuscript describes a work which is seriously and rigorously performed. The simulations are analysed thoroughly with a serious quantification effort. The authors have made an important effort of clarity in their study.

My central question is about the motivation of the manuscript. The central theme of the manuscript authors study « contrarian » active T1 transitions which induce convergence extension in the direction orthogonal to an externally applied stress. I would disagree with the authors that this is the only type of active T1 transition. For instance polarised transitions which are oriented by the epithelium chemical polarity and induce converge-extension in the absence of external tension, as is thought to be the case during germ-band elongation, could also be called « active ». From the manuscript it is not clear to me why inducing convergence extension as a response to external tension, against the external tension, is important or biologically relevant. I think the authors should significantly clarify the rationale for their study.

In that respect I am not sure that the comparison with experiment is very useful. My understanding is that essentially myosin cables in the chick embryos appear oriented along the direction of tissue contraction, which seems to make sense if these cables are actively contracting. I am not sure that this observation strongly supports the much more detailed model proposed by the authors; notably the notion that myosin polarises in response to external tension.

Judging from Figure 6, the authors also observe relatively limited convergence extension happening against the external force. I wonder if this may be due to myosin molecules playing both the role of the force-exerting molecule and the sensor in the model? For instance, possibly if a secondary molecule responding to external tension, and itself inducing myosin polarisation, a more permanent convergence-extension could be induced?

– I think the authors may have overlooked the reference « active instability and nonlinear dynamics of cell-cell junctions », Krajnc, et al, PRL, 2021. The approach and questions seem very close to the current manuscript. I think the results and approach of this study should be contrasted to the current work.

– When comparing Figure 7C and Figure 6, I had the impression that Figure 7C showed significant « negative » convergence extension for large values of β (0.75), why the authors say that this is not the case in the caption of Figure 6?

– The model assumes a constant number of available myosin molecules in each cell. It was not clear to me if this assumption is important for the results of the model?

– I do not understand the reason for the factor « z » in the second term of the right-hand side of Equation 5.

*Reviewer #3 (Recommendations for the authors):*

The authors propose a mechanism for the dynamics of cell-cell junctions under an applied external mechanical load. In their theoretical analysis they assume that myosin recruitment and therefore active contractile stress in a junction increases with an applied external load. They find that this mechanism can induce a T1 transition through which adjacent cells exchange neighbours such that an originally aligned junction is reorientated perpendicularly to an applied uniaxial extensile stress. By using a generalised vertex model they also find that this effect on an isolated cell-cell junction can lead to convergence-extension of a tissue, where the extension occurs perpendicularly to the applied uniaxial extensile stress. In both cases, the phenomenon is present for a broad range of parameter values. Eventually, the authors compare their theoretical results to convergence extension of a part of a tissue in an early chick embryo.

The theoretical analysis is carried out mostly through simulations and is done in a convincing way. The dynamics assumed for a single junction is based mostly on reasonable assumptions and differs from earlier works. It appears though that the assumption of a monotonic decrease of the myosin unbinding rate with the applied force needs more justification. I would assume that for sufficiently high forces, this rate will increase as is the case for catch bonds. Taking such an effect into account might limit the range of parameter values for which stress-induced convergence-extension is observed. Also, it was unclear to me whether all ingredients of the model where necessary to obtain the observed effect. In particular, I wonder what happens if either of the dashpots or the springs are eliminated? Finally, I do not understand why m_0 should describe effects of the surrounding. Shouldn't this effect be included in T_ext? Otherwise the theoretical analysis is convincing and justifies the conclusions of the authors.

I am less convinced by the comparison with experiments on gastrulation in early-stage chick embryos. Although the relative orientation between myosin cables and the direction of extension agrees with that in the theory, I fail to see where the external uniaxial extensile stress should come from. It thus appears that an essential element of the theory is not checked for in the experiments. The authors write themselves in ll 395: "Although it is yet to be experimentally confirmed, it is plausible that the symmetry breaking event that induces the initial myosin polarity occurs as a result of anisotropic tension combined with cell differentiation early in development." The authors should clarify this point, which seems to be central to the authors' idea that mechanical signalling drives convergence-extension in this case. Otherwise, the experimental data do not add much to the work.

The authors frequently use the term "polarisation" or "polarised" when referring to the alignment of the myosin cables, for example. This notion implies directionality, for example, a difference between left and right in Figure 2, which is not present. I would suggest that the authors rather use 'anisotropy'.

The text exhibits some jumps between figures that are not always referred to in order, for example between Figures2 and 3 or Figure 4C and 4B. The authors might want to adapt their text to the figures or vice versa.

In Figure 1A the term \β m is hard to see. Please, improve.

In Figure 3 the colour code is confusing. In A black, red, blue correspond to different junction classes, in B and C to different mechanical quantities and in D again to different junction classes but not the ones in A. Please, improve.

Ll 49 "With cellular behaviours being coordinated over thousands of cells in the case of the chick embryo, biochemical signalling alone is unlikely to account for the observed motion patterns." Even though I have a guess, I do not really understand this argument. Can you add some words to explain, why the long-range coordination is at odds with pure biochemical signalling?

L 218 side a -> side OF a

L 256 along the direction the central junction -> along the direction OF the central junction

Ll 274 could you add a panel to show this?

In Figures6 and 7A, convergence-extension is not really visible assuming that the initial state was square. Maybe you can show a sequence similar to Figure 2?

[Editors' note: further revisions were suggested prior to acceptance, as described below.]

Thank you for resubmitting your work entitled "Generating active T1 transitions through mechanochemical feedback" for further consideration by *eLife*. Your revised article has been evaluated by Aleksandra Walczak (Senior Editor) and a Reviewing Editor.

The manuscript has been improved but there are some remaining issues that need to be addressed, as outlined below:

Your model doesn't seem to apply to the fly germband elongation, in which the invaginating midgut (pulling external force) and the T1s occur along the same axis, with cables perpendicular to the pulling axis. In your response, you insist that the pulling force is rather the ventral invagination than the posterior midgut (hence a force perpendicular to tissue elongation and T1s, and parallel to the cables, as in their model). This overlooks that the germband does not extend without the posterior midgut pulling (Torso), while it does without the ventral mesoderm invagination. In the paper, claims about their finding applying to germband extension are found in the intro and discussion, without other justification. These comments should be removed unless very strongly justified, as it seems that they are based on a direct misinterpretation of previous observations.

When the germband is mentioned you usually cite a paper that does NOT deal with the germband (Jacinto et al. 2002, which is about dorsal closure). And you also cite Duda et al., even though it's a wing disc paper, in which cables form parallel to the pulling force, (as in their model, but in contrast with the germband). This could be confusing and even detrimental to your work when readers familiar with fly morphogenesis read it.

---

## [Author Response]

Essential revisions:1) Improve the comparison with experiments (see reports).

In the introduction of the revised manuscript, we have highlighted the experimental data on large scale tissue flows and the underlying oriented myosin driven intercalations observed in the chick embryo. These experiential observations directly inspired the tension-sensitive myosin recruitment model studied in this work. Furthermore, we have included the discussion about the catch-bond mechanism that has here been proposed as a simple mechanism of tension dependent myosin accumulation. We, however, also explicitly discuss that the critical requirement for this model to lead to active contraction is tension sensitive myosin accumulation and that mechanisms other than the proposed catch bond mechanism could be involved. Resolving the actual mechanism in detail is, however, beyond the scope of this work, and will have to await further experiments.

2) Argue more convincingly for biological relevance.

We have extended the explanation for the biological relevance for tension-dependent myosin mediated control of the direction and patterns of cell intercalations. This explanation is primarily based on the observation that the intercalations need to be organised robustly over very large (several millimetres) domains. In chicken embryos, there is a clear spatial pattern of propagation of the onset of motion, which suggests a rapidly propagating signal as well as the dynamic patterns of cell deformations that we have observed. Those correlate closely with the formation and direction of supercellular myosin cables.

3) Put the positive feedback tension/myosin more into context and discuss possible effects of mechanisms that are an alternative to the one you implemented (see reports).

In the revised manuscript, we discuss the catch bond idea in more detail and mention explicitly that there could be additional and/or other molecular mechanisms of implementing a tension dependent myosin recruitment.

Reviewer #1 (Recommendations for the authors):This paper presents a numerical study of tissue convergent extension during epithelial morphogenesis. It builds upon several earlier works, including so-called vertex models, and single junction mechanical models. The specificity of the model presented lies in its active component: a positive feedback exists between Myosin activity and mechanical tension in each junction. This leads to interesting tissue-scale dynamics. Upon tissue pulling, the authors report conditions for which intercalation occurs perpendicular to the pulling direction. They provide a rather detailed analysis of the model and its numerical outcome, and a qualitative comparison with convergent extension in avian embryos.

We thank the review for their insightful comments on our work. Below, we address the specific issues raised by the reviewer.

Strengths:The model is presented in a rigorous manner and with a lot of details and pedagogy. It is discussed in comparison with other models of the field, and the similarities and differences are clearly and honestly presented by the authors in their discussion.The numerical study provides interesting and possibly counter-intuitive predictions in which tissue convergent extension can occur perpendicular to an external pulling axis, due to a positive feedback between myosin motors recruitment and tension which leads to the formation of Myosin chains (or cables) along the pulling axis.The conditions in which this regime of tissue remodeling can occur (under the hypotheses of the model) are thoroughly investigated. Finally the authors go a bit beyond the sole numerical study, and analyze experimental data of the polarized tissue flow occurring upon primitive streak formation in avians.Weaknesses:Although the numerical investigation is thorough and the model hypotheses are clearly stated, the paper falls a bit short of convincing readers that the emergent behaviors observed in the model are of actual biological relevance.

In the introduction of the revised manuscript, we have extended the discussion about experimental observations of supercellular myosin cables that are believed to drive contractions and active cell intercalation events in early-stage chick embryos. These experiments have been the direct motivation for the model. The model, however, does not aim to be a specific, detailed description of the chick embryo, but instead aims to explore whether a relatively simple and generic mechanism of tension-controlled activation of myosin motors can lead to the observed convergence-extension tissue deformations.

The hypotheses and ingredients of the model are clearly stated (1) junctions are viscoelastic, (2) junctions are active, i.e. they can generate tension through myosin, and (3) positive feedback between tension and myosin. Yet the 3rd hypothesis, which gives its singularity to the model, is not discussed enough, and its relevance to well-known convergent extension systems is not analyzed. In the paper, the positive feedback arises from the decrease of unbinding rate upon tension increase. This "catch bond" hypothesis (note: the term catch-bond only appears once, in the methods) is at the core of all the results obtained in the paper, and this should be clearly stated and discussed. As for the positive feedback, one could also imagine that it is not the unbinding rate that decreases upon tension increase, but rather the myosin recruitment rate that increases. Would that be formally equivalent? The authors should address this point and its biological implications.The comparison between the model outcome (notably, the T1 transitions preferably oriented in the direction perpendicular to pulling) and existing experimental data is a bit elusive. The authors propose a semi-quantitative comparison with convergent-extension occurring upon primitive streak formation in avians. The data and conclusion is not fully convincing, partly because of the polarization data itself (Figure 8C), and partly because it's not clear what would be the equivalent of the external pulling force in this self-polarizing flow.The authors also mention the germband of , but the model is, I believe, not relevant for that system. Although there is indeed an external force pulling the tissue (pulling posterior midgut), polarization is perpendicular and extension parallel to the pulling force (the opposite of the model outcome).

We hope that the revised manuscript now makes it clear that there may be external forces as exerted by the *Drosophila* midgut invagination during germband extension, but the idea for the chick embryo is that the intercalations start somewhere in the tissue (e.g. possibly by localised ingressions of cells in the sickle region). These intercalations exert mechanical stresses on neighbouring cells, which develop anisotropy themselves, contract, and activate their neighbours, leading to a chain-like sequence of intercalations. This is similar to the *Drosophila* case, where the apical contraction of the mesoderm precursor cells in the ventral midline resulting in their invagination in the ventral furrow, exert a pulling force on the lateral mesoderm. Also, in this case that cables formed are oriented towards the ventral furrow, resulting in contraction of this tissue along the direction of the myosin cables and extension perpendicular to those.

In that spirit, they should also discuss pulling experiments (fly wing disc / Mao lab, suspended epithelia / Charras lab) where it's not so clear that topological transitions occur upon pulling.

In the revised manuscript, we have pointed out that the fly wing disk of the Mao lab and the suspended epithelia of the Charras lab are epithelia likely made up of cells that are in a different more mature differentiation state and may not be able to undergo the same amount of intercalations as the mesendoderm precursor tissues in the chick and the fly. Neither pulling experiments on cultured MDCK epithelial cell monolayers (Harris, et al. 2012) nor on the fly wing disk (Duda, et al. 2019) show any topological transitions even at stains above 50%. In the case of the MDCK monolayers, it is attributed to the slow turnover of keratins and desmosomes (~1h), reflecting their more mature differentiation state. Therefore, while it is possible that topological transitions would occurs over long times, the mechanism presented in this work would not likely apply to stretching experiments of these tissues.

Finally, the paper builds upon already existing models of junction and tissue mechanics. Beyond the vertex model itself, the hypothesis of a feedback between myosin activity and tension has also been around in the field for some time, including in modeling papers (which authors acknowledge honestly).In conclusion:The paper presents a thorough numerical analysis of T1s and convergent extension upon tissue pulling when a positive feedback exists between Myosin and junction tension. Yet, the fact that this type of model and hypotheses are not completely new, combined to a lack of comparison to experimental data (except for avian gastrulation) makes it a bit difficult for the reader to assess the novelty of the findings presented in the paper.I liked the presentation of the model a lot, the details given, the pedagogy, the comparison with what already existed (Discussion section), and the thorough analysis of the model outcome.On the other hand, I find it a bit difficult to see whether the novelty is strong enough for publication in eLife:First, because somewhat similar models exist in the field. According to the authors, the specificity here is the inclusion of the positive feedback for the active part of the model. Yet, other models that the authors cite include a similar type of feedback, although they might not specifically focus on topological transitions and convergent extension. In addition, the existence of such a feedback (and its possible consequences on tissue morphogenesis) has been around in the field for a while, under various forms (here, the authors chose a catch-bond type dissociation rate for myosin).

We agree with the reviewer that similar models already exist in the literature. There is growing evidence that many tissue-scale processes are driven by myosin activity that manifests itself as active contractions and expansions of cell-cell junctions that lead to cell rearrangements, tissue ordering, etc. Assuming some form of tension-dependent myosin dynamics (e.g. catch-bonds), it is not surprising that many authors have developed similar models.

It is, however, surprising that models that share common basic principles can capture, at least qualitatively, rather different biological processes in different organisms. This suggests a set of universal physical processes that govern tissue scale behaviours.

In this work, we have applied a version of a model that treats cell-cell junctions as active entities to the previously unexplored problem of T1 transitions that occur against externally applied tension, a process of importance in understanding early development of avian embryos. We, therefore, argue that this works makes a significant contribution to the existing literature of understanding the role of activity in epithelial tissue mechanics. In addition, our computational model is able to treat the whole sequence from the application of external forces to the induced tissue flow through T1 transitions, i.e. the full emergent active rheology, a significant advance compared to other models.

In the revised manuscript, we make the connection to other models clearer.

Second, because the comparison (even qualitative) with existing experimental data remains elusive.Hence the major finding of this paper is that a tissue in which a positive feedback between tension and motor recruitment exists can display active intercalations and convergent extension perpendicular to an external pulling force. In which experimental systems this prediction could be relevant remains to be determined.

In the revised manuscript, we make this connection clearer by referring to experiments on primitive streak formation in chick embryos. Current state of the art experiments, however, cannot yet follow myosin dynamics in-vivo, and that is why we focus on cell intercalations that, we argue, are a good readout of myosin driven active contractions. We also reiterate that the other main qualitative evidence for the tension feedback state, tissue extension is orthogonal to the main direction of cell elongation, is seen in both model (Figure 7C) and experiment (Figure 8D-E). We emphasise, however, that the aim of this work is to study a generic biologically plausible model for tension-driven activation of cell-cell junctions, and its effects on cellular behaviours.

– Does the positive feedback tension/myosin need to stem from the catch bond model? Could one imagine increased motor recruitment open junction/cell stretch (This would be in line with some experimental data, including stretching experiments). Would that be formally equivalent to the catch bond, or would that change the outcome of the model? If yes, how?

While the proposed model is coarse and does not make any specific assumptions about the molecular mechanisms behind the positive tension/myosin feedback, the catch-bond mechanism would be a plausible molecular realisation for the proposed model. The assumption that the myosin association rate is tension independent and that the dissociation rate decays with tensions is based on experiments of Kovács et al. (PNAS 104, 9994, 2007) on non-muscle myosin II that is believed to be the key molecular motor for active processes studied in this work.

Due to the coarse-grained nature of the model, assuming that motor recruitment depends on tension would not lead to qualitatively different results compared to the current model. The reason is that the central assumption is that external tension leads to accumulation of motors and, hence, increased contractility, without focusing on the specific molecular origin of such build-up.

We have revised the manuscript to make this point clearer.

– In which systems do people observe tissue extension along the direction perpendicular to pulling? The authors only refer to the avian embryo in which the nature of the flow (embryo-scale) makes it a bit difficult (at least for me) to interpret what is the chicken and what is the egg. A lot of data is now available out there for analysis. It would be great to see the same kind of analyses in other experimental systems. Notably, the fly germband (that the authors rapidly mention) seems to behave completely differently, with cables of Myosin perpendicular to the pulling axis (pulling force being exerted by the posterior midgut). How about convergent extension in the zebrafish or *Xenopus* embryos?

This model has been directly inspired by experiments on avian (chicken, to be precise) embryos, where there has been no clear evidence of chemical pre-patterning. The model analysed in this study proposed a plausible cell-level mechanism for observed cell intercalation events that occur along the direction of the actomyosin cables, believed to be a proxy for the direction of mechanical tension. This mechanism is also applicable to the germband extension in the fly, where the apical contraction of the cells in the ventral midline also exert pulling forces on the lateral mesoderm, resulting in myosin cables driving junction contractions of intercalating cells towards the ventral midline.

We briefly note that direct comparison with *Xenopus* and the zebrafish would not be possible. In those organisms, the morphogenic flows involve mesenchymal cells (not epithelial cells) and they do exhibit a different mechanism of intercalation movement, i.e. their migrate on top of each other, which is different from junction shortening and lengthening. It still involves polarisation, extension of cellular process and myosin II driven contraction. The detailed mechanism is, however, less clear in this case (Pfister, et al., *Development* 143.4, 715, 2016).

– What happens at the boundaries? Here we look at a central patch of cells. And somewhat related question, how would that work in a closed geometry (such as the fly embryo). I feel that the avian discussion (paragraph 1 of discussion) could be expanded with considerations on how the feedback can produce long range flows even though only a small patch of cells is initially "activated".

The boundaries could indeed play an important role, especially in the case, as pointed out by the reviewer, of a closed geometry such as the embryo. This is clearly the case both for the fly and the chick embryo, for which in the early stages of the development, the central embryonic area is surrounded by the area opaca (a thicker region of cells attached to the vitelline membrane) that provides tension to the central region. Our attempts to simulate such geometries (not shown) led to unsatisfactory results. The posterior sickle-shaped active region indeed starts to contract and exhibits motion reminiscent of convergence-extension but loses coherence and does not lead to a streak-like structure. We speculate that presence of cell divisions and cell ingressions plays an important role in large scale flows (as suggested by recent experiments Chuai, et al. *Science Advances* 9.1, eabn5429, 2023), which were not included in the model. One major effect of division and ingression is that the tissue is always a fluid without a yield stress (Matoz-Fernandez, et al. Soft Matter 13, 3205, 2017), which will indeed facilitate activity-induced flows. Such effects will, however, be explored in detail elsewhere. The goal of this study is primarily to showcase that under simple assumption of tension dependent myosin kinetics, it is possible to have cell intercalations that are perpendicular to the direction of external tension.

– Authors should cite Duda et al. (Dev Cell 2019) from the Mao group. In this article it is shown that Myosin polarizes and form cables upon external tissue stretch in the fly wing disc. This is fully relevant to their model and it is somewhat surprising that they do not cite it. Disclaimer: I'm not on that paper

We thank the reviewer for pointing out the Duda, et al. paper. It is indeed very relevant for our work, and it was omitted by from the original submission by mistake. We refer to it in the revised manuscript.

– The authors present their model as "one of the simplest descriptions" that features the three main ingredients required (viscoelastic junctions, active junctions, positive feedback myosin/tension). What would be the simplest and why didn't they choose it?

This is statement has been poorly worded and it is meant to convey that we have chosen a minimal model that is able to describe a cell-cell junction that has a viscoelastic, an elastic and an active component, and acts against the elastic background of the surrounding tissue that is under tension. In the revised manuscript, we have rephrased the sentence to read: “In order to describe the three key ingredients that characterise a cell-cell junction, while taking into account the effects of the elastic background, we adopt a minimal description, i.e. we assume that the junction consists of four elements connected in parallel:”.

– Authors should better explain the rationale of the "elastic barrier" term B(l-a) , as compared to the k(l-l0) term.

The *k*(*l* – *l*_0_) term describes the viscoelastic properties of the junction itself, which is encoded in the time-dependent value of rest length *l*_0_ that relaxes towards the actual junction length. The elastic barrier, on the other hand, refers to the presence of the surrounding tissue that is assumed to be under tension and therefore, its response contains a purely elastic component, which acts against junction remodelling. In the language of soft matter physics, it represents the yield stress of the surrounding tissue. Clearly, there are multiple possible approaches to modelling the effects of such elastic background. We opted for a simplistic description that models the elastic background as the additional elastic spring of stiffness B. While this is a rather crude approximation, it is reasonable to assume that more sophisticated approaches would not qualitatively change the results. In the revised manuscript we have add the following sentences: “Furthermore, epithelial tissues, and early-stage embryos, in particular, are often under mechanical tension. This tension generates an elastic background against which the junction contracts and expands. We refer to that background elasticity as the *elastic barrier* since it is assumed that it acts to prevent junction remodelling. ”

Reviewer #2 (Recommendations for the authors):Sknepnek et al. study a model for junction dynamics in a two-dimensional vertex model, intended to describe epithelial mechanics. In the model, a concentration of myosin on each half-junction evolves according to a chemical balance equation and induces an active tension in the junction. The chemical balance equation incorporates a mechanical feedback, whereby myosin unbinding is inhibited at high tension. The authors first study a single junction and consider thresholds for the junction to fully contract. They then investigate the behaviour of a few « active » junctions in a passive network and a fully active vertex model. They find that at intermediate external pulling force and myosin-induced tension, the simulated epithelium can undergo convergence-extension against the pulling force.The manuscript describes a work which is seriously and rigorously performed. The simulations are analysed thoroughly with a serious quantification effort. The authors have made an important effort of clarity in their study.

We thank the reviewer for insightful comments and suggestions. We address each point below.

My central question is about the motivation of the manuscript. The central theme of the manuscript authors study « contrarian » active T1 transitions which induce convergence extension in the direction orthogonal to an externally applied stress. I would disagree with the authors that this is the only type of active T1 transition. For instance polarised transitions which are oriented by the epithelium chemical polarity and induce converge-extension in the absence of external tension, as is thought to be the case during germ-band elongation, could also be called « active ». From the manuscript it is not clear to me why inducing convergence extension as a response to external tension, against the external tension, is important or biologically relevant. I think the authors should significantly clarify the rationale for their study.

We agree with the reviewer that T1 transitions studied here are not the only possible type of active T1 transitions. In the revised manuscript, we have added the following sentences: “The focus of this work is on the T1 transition that occurs perpendicular to the direction of the maximum principal mechanical stress. We refer to such T1 transition as *active*, and remark that other types of active T1 transition in embryonic tissues are possible, e.g. polarised transitions oriented by the epithelium chemical polarity that induce converge-extension in the absence of external tension, as is thought to be the case during germ-band elongation (Bertet, et al. (2004)).”

In that respect I am not sure that the comparison with experiment is very useful. My understanding is that essentially myosin cables in the chick embryos appear oriented along the direction of tissue contraction, which seems to make sense if these cables are actively contracting. I am not sure that this observation strongly supports the much more detailed model proposed by the authors; notably the notion that myosin polarises in response to external tension.

The main idea is that some cells start to contract and then pull on neighbouring cells as discussed in the revised text, resulting in myosin accumulation and building up of supercellular chains that then, most likely, direct the intercalation chains. The chains of intercalations that we see in the full vertex simulations have been shown to exist in the experiments.

Judging from Figure 6, the authors also observe relatively limited convergence extension happening against the external force. I wonder if this may be due to myosin molecules playing both the role of the force-exerting molecule and the sensor in the model? For instance, possibly if a secondary molecule responding to external tension, and itself inducing myosin polarisation, a more permanent convergence-extension could be induced?

The reviewer raises and interesting and important question, which, unfortunately, cannot be directly answered with the current model. We speculate that a more prominent convergence extension would require spatial and temporal coordination of active T1 events. Such coordination does not seem to arise in the current model, and we speculate that it is possibly related to the effects the reviewer has alluded to. For example, in the current model, the post-T1 expansion of the newly formed junction is effectively passive. Additionally, our model omits divisions and ingressions which necessarily induce flow in model tissues (Matoz-Fernandez, et al. Soft Matter 13, 3205, 2017) and could help break “stuck” myosin cables that halt convergence-extension flows in the simulations.

To provide a detailed answer to reviewer’s question, however, one would need to extend the part of the model that describes kinetics of myosin motors to include additional species of molecules. Such efforts have been made, albeit in a different context, e.g. by Lan, et al. (2015). It would also likely be necessary to modify the coupling between myosin kinetics and mechanical tensions. For example, recent studies by Krajnc, et al. (2021) and PérezVerdugo and Banerjee (arXiv preprint, Nov 2022) suggest that tensions fluctuations have interesting effects on the cell intercalation dynamics, among other things leading to a nonintuitive observation of enhanced occurrences of T1 transitions with increasing mechanical tension. Extending the model to include such effects, while very interesting, is beyond the scope of this work, and will be addressed elsewhere.

– I think the authors may have overlooked the reference « active instability and nonlinear dynamics of cell-cell junctions », Krajnc, et al, PRL, 2021. The approach and questions seem very close to the current manuscript. I think the results and approach of this study should be contrasted to the current work.

We thank the reviewer for pointing out the paper by Krajnc, et al. 2021. In the revised manuscript, we cite it and comment on its main conclusions.

– When comparing Figure 7C and Figure 6, I had the impression that Figure 7C showed significant « negative » convergence extension for large values of β (0.75), why the authors say that this is not the case in the caption of Figure 6?

The reason convergence extension in Figure 6 appears less significant compared to the measurements shown in Figure 7C, is due visualisation in Figure 6. In simulations, the activity was first switched off and the system was first stretched until it reached the mechanical equilibrium. The activity was then switched on and the system was allowed to actively contract. The amount of convergence-extension was measured relative to the initial stretched state when the activity was first switched on. In order to make this clear, in the revised manuscript, we have included a short supplement video to Figure 6 that shows the initial passive stretching and subsequent active contractions for the same parameters used in Figure 6.

– The model assumes a constant number of available myosin molecules in each cell. It was not clear to me if this assumption is important for the results of the model?

Conservation of myosin per cell is introduced in order provide a mechanism that would allow coupling of myosin distributions on neighbouring junctions. In other words, the assumption is that junctions adjacent to a given junction act to deplete its myosin, which is an important assumption required for the proposed mechanism to drive an active T1 transition. Clearly, there are many ways to model such mechanism and we explored several possible choices. We settled with the conserved myosin model since it has only one free parameter (i.e. the total myosin per cell), while being plausible. In the revised manuscript, we have added the following sentence to make this point clear: “The finite pool of myosin acts to introduce correlations in the distribution of junctional myosin within each cell. In other words, it models the mechanism by which, e.g. depletion of myosin on a given junction is correlated to myosin accumulation on its neighbouring junctions. This is of central importance to establish myosin anisotropy within the cell. There are clearly many alternative ways to model coupling of myosin on different junctions within a cell. The model used here, however, requires a minimal number of parameters while being biologically plausible.”

– I do not understand the reason for the factor « z » in the second term of the right-hand side of Equation 5.

The factor “z” in Equation 5 is the number of junctions of a given cell. It is there because of the assumption of conserved myosin in each cell as discussed in the answer to the reviewer’s previous question. Its origin is explained in the discussion leading to Equation 12. Its presence, however, does not qualitatively affect the behaviour of the model, and even in the case of a random tissue patch, without the loss of generality, one can set z=M, where M is the total myosin pool in the cell.

Reviewer #3 (Recommendations for the authors):The authors propose a mechanism for the dynamics of cell-cell junctions under an applied external mechanical load. In their theoretical analysis they assume that myosin recruitment and therefore active contractile stress in a junction increases with an applied external load. They find that this mechanism can induce a T1 transition through which adjacent cells exchange neighbours such that an originally aligned junction is reorientated perpendicularly to an applied uniaxial extensile stress. By using a generalised vertex model they also find that this effect on an isolated cell-cell junction can lead to convergence-extension of a tissue, where the extension occurs perpendicularly to the applied uniaxial extensile stress. In both cases, the phenomenon is present for a broad range of parameter values. Eventually, the authors compare their theoretical results to convergence extension of a part of a tissue in an early chick embryo.

We thank the reviewer for critical reading of our manuscript and helpful and insightful comments and suggestions.

The theoretical analysis is carried out mostly through simulations and is done in a convincing way. The dynamics assumed for a single junction is based mostly on reasonable assumptions and differs from earlier works. It appears though that the assumption of a monotonic decrease of the myosin unbinding rate with the applied force needs more justification. I would assume that for sufficiently high forces, this rate will increase as is the case for catch bonds. Taking such an effect into account might limit the range of parameter values for which stress-induced convergence-extension is observed. Also, it was unclear to me whether all ingredients of the model where necessary to obtain the observed effect. In particular, I wonder what happens if either of the dashpots or the springs are eliminated? Finally, I do not understand why m_0 should describe effects of the surrounding. Shouldn't this effect be included in T_ext? Otherwise the theoretical analysis is convincing and justifies the conclusions of the authors.

As shown in Figure 4A, there is indeed a finite range of parameters over which the convergence-extensions occurs. This is because at very high applied tensions, passive T1 transitions in the direction of pulling additionally appear. We agree that effects such as the increase of the detachment rate at very high tensions would indeed affect the range of convergence-extension, by contributing to another increase in passive T1 transitions at very high stresses. We argue, however, that the parameter range favourable for convergence extension is still sufficiently wide to be biologically plausible. Additionally, the tissue eventually extending in the pulling direction at extreme forces is also realistic.

Regarding the other ingredients, in the revised manuscript, we explain the motivation behind and the significance of each element in the model. We argue that this is the smallest number of elements that are necessary to observe tension-driven activation and subsequent convergence-extension flows. Therefore, eliminating any of the elements would lead to qualitative changes in its behaviour. The fact that we need a second spring, *m*_0_ (i.e. the myosin reference level) and *T** (i.e. the threshold tension) is a peculiarity of the vertex model. A large number of iterations over the details of the vertex model showed that the perimeter term (responsible for the second spring) is necessary to retain polygonal cell shapes. This induces self-stresses in the tissue, requiring a *T** so that junctions do not randomly activate. Equally, for the active parts if we, e.g. implement a shift by setting both *T** and *m*_0_ to 0, individual junctions in the tissue activate spontaneously at no applied force. Results from a continuum model (to be published separately) indicate that in the absence of the local self-stresses induced by a disordered vertex model, *T** and the second spring can indeed be omitted.

I am less convinced by the comparison with experiments on gastrulation in early-stage chick embryos. Although the relative orientation between myosin cables and the direction of extension agrees with that in the theory, I fail to see where the external uniaxial extensile stress should come from. It thus appears that an essential element of the theory is not checked for in the experiments. The authors write themselves in ll 395: "Although it is yet to be experimentally confirmed, it is plausible that the symmetry breaking event that induces the initial myosin polarity occurs as a result of anisotropic tension combined with cell differentiation early in development." The authors should clarify this point, which seems to be central to the authors' idea that mechanical signalling drives convergence-extension in this case. Otherwise, the experimental data do not add much to the work.

In the revised manuscript, we make the connection with the experiments clearer. Namely, we point out that mechanism does not need specific external forces, and the pulling forces can be generated within the tissue as described several times in the text and above. In order to avoid the complicated problem of the origin of pulling forces in real biological system (which is also beyond the scope of this work), in the model we used external forces solely for the purpose of generating the initial anisotropy that activates the system. We have also added an example of a central myosin pulse triggering a chain contraction in a one-dimensional system (new Figure 1 – supplement figure 2).

The authors frequently use the term "polarisation" or "polarised" when referring to the alignment of the myosin cables, for example. This notion implies directionality, for example, a difference between left and right in Figure 2, which is not present. I would suggest that the authors rather use 'anisotropy'.

We agree with the reviewer that the term “polarisation” might be confusing. It indeed is meant to mean “anisotropy”. In the revised manuscript, we have used the term “anisotropy” instead.

The text exhibits some jumps between figures that are not always referred to in order, for example between Figures2 and 3 or Figure 4C and 4B. The authors might want to adapt their text to the figures or vice versa.

We have updated Figure 2 to have only two panels (A and B). We have swapped panels B and C in Figure 4, so they are no referenced in order in the text.

In Figure 1A the term \β m is hard to see. Please, improve.

We have updated Figure 1A such that β*m* is clearly visible.

In Figure 3 the colour code is confusing. In A black, red, blue correspond to different junction classes, in B and C to different mechanical quantities and in D again to different junction classes but not the ones in A. Please, improve.

We have modified colours in Figure 3 and Figure 2B —figure supplement, and updated figure captions, such that there is no longer ambiguity about what each colour represents.

Ll 49 "With cellular behaviours being coordinated over thousands of cells in the case of the chick embryo, biochemical signalling alone is unlikely to account for the observed motion patterns." Even though I have a guess, I do not really understand this argument. Can you add some words to explain, why the long-range coordination is at odds with pure biochemical signalling?

Long range biochemical signalling requires either setting up and maintaining gradients of signalling molecules, which is a slow process over larger distances. These gradients would be deformed continuously by the tissue flows making it difficult to see how this would be maintained. Alternatively, some kind of small molecule cell to cell signal relay mechanism, such as the cAMP relay mechanism that controls the chemotactic aggregation of *Dictyostelium* cells, could be present. However, so far, no evidence for any kind of long-range relay mechanism exists. It seems much more logical that stresses generated locally due to specific cell behaviours, and which spread rapidly and over longer distances into surrounding tissues, influence directly biochemical signalling via mechanosensitive biochemical reactions. These have now been abundantly demonstrated to exist, especially mechanical signalling to the cytoskeleton that directly affect cellular function and behaviours such as adhesion and motility.

L 218 side a -> side OF a

Fixed.

L 256 along the direction the central junction -> along the direction OF the central junction

Fixed.

Ll 274 could you add a panel to show this?

The behaviour is very similar to what is shown in Figure 7 —figure supplement 3B for the random patch. The regime where this happens is not biologically relevant and the model tissue made of hexagonal cell only is rather artificial. Therefore, we do not believe that adding a panel would be warranted. In the revised manuscript, we, however, alert the reader to the discussion in the section devoted to a fully random active patch.

In Figures6 and 7A, convergence-extension is not really visible assuming that the initial state was square. Maybe you can show a sequence similar to Figure 2?

We agree with the reviewer that the static images do convey convincingly convey the extent of convergence-extension. To address this, in the revised manuscript, we added a video supplement to Figure 6 that shows the initial passive stretching and subsequent active contractions, thus showing the full convergence-extension process.

[Editors' note: further revisions were suggested prior to acceptance, as described below.]

The manuscript has been improved but there are some remaining issues that need to be addressed, as outlined below:Your model doesn't seem to apply to the fly germband elongation, in which the invaginating midgut (pulling external force) and the T1s occur along the same axis, with cables perpendicular to the pulling axis. In your response, you insist that the pulling force is rather the ventral invagination than the posterior midgut (hence a force perpendicular to tissue elongation and T1s, and parallel to the cables, as in their model). This overlooks that the germband does not extend without the posterior midgut pulling (Torso), while it does without the ventral mesoderm invagination. In the paper, claims about their finding applying to germband extension are found in the intro and discussion, without other justification. These comments should be removed unless very strongly justified, as it seems that they are based on a direct misinterpretation of previous observations.

We thank the reviewers and editors for this comment. We agree that extrinsic forces associated with midgut invagination are involved in germband extension and that in the torso mutant there is no germband extension. However, in the original experiments by Irvine and Wieschaus (1994) [doi.org/10.1242/dev.120.4.827] they state that although there is no posterior movement, cell intercalations do not stop. More recent experiments by Collinet, et al. (2015) [doi.org/10.1038/ncb3226] show that both in Torso mutants as well as in embryos where a DV line of tissue in the posterior of the embryo is cauterised to the eggshell, cell intercalations and junctional myosin accumulation continue, showing that intercalations are either autonomous or dependent on other external forces. Furthermore, the same work shows via junction ablation experiments that the tension in the DV junctions is higher than in AP junctions both in wildtype as well as in Torso mutants. The DV/AP aspect ratio of the cells is initially 1.4 in the wildtype decreases to smaller than 1 over time, while in Torso mutants it stays close to 1.2. This would be in line with the observation that at least initially the DV tension dominates over the AP tension. More recently experiments by Gustafson, et al. (2022) [doi.org/10.1038/s41467-022-34518-9] have shown that the strain rate shows a DV gradient with highest value that ventral side and this correlates strongly with the junctional myosin association rate. Furthermore, they also showed that contraction of cells using optogenetic techniques can induce deformation and myosin accumulation in neighbouring cells. In twist and snail mutants where no ventral furrow forms, germband extension is greatly slowed down, again hinting at the importance of that process in driving intercalations. Furthermore, Lefebvre et al., (2023) [doi.org/10.7554/*eLife*.78787] have shown that during germband extension the orientation of the myosin anisotropy remains primarily DV oriented, while the pair rule gene expression domains show a high vorticity, suggesting an indirect or complex coupling of gene expression to myosin localisation. Therefore, we think that the type of feedback we propose in the model, together with morphogenetic patterning, is relevant to the process of germband extension. We have rewritten the section on *Drosophila* germband extension to make this more explicit. The updated text is shown in magenta.

When the germband is mentioned you usually cite a paper that does NOT deal with the germband (Jacinto et al. 2002, which is about dorsal closure). And you also cite Duda et al., even though it's a wing disc paper, in which cables form parallel to the pulling force, (as in their model, but in contrast with the germband). This could be confusing and even detrimental to your work when readers familiar with fly morphogenesis read it.

We completely agree with these comments that this was not clearly explained. The Jacinto paper was cited as another case where supercellular myosin cables were observed, while the Duda paper was cited as another case where it as been shown that tension can result in myosin recruitment. These observations, although relevant, did not relate to germband extension and we have removed them from the revised version.